# ST-PPO: Stabilized Off-Policy Proximal Policy Optimization for Multi-Turn Agents

## Abstract

Proximal policy optimization (PPO) has been widely adopted for training large language models (LLMs) at the token level in multi-turn dialogue and reasoning tasks. However, its performance is often unstable and prone to collapse. Through empirical analysis, we identify two main sources of instability in this setting: (1) token-level importance sampling, which is misaligned with the natural granularity of multi-turn environments that have distinct turn-level stages, and (2) inaccurate advantage estimates from off-policy samples, where the critic has not learned to evaluate certain state-action pairs, resulting in high-variance gradients and unstable updates. To address these challenges, we introduce two complementary stabilization techniques: (1) turn-level importance sampling, which aligns optimization with the natural structure of multi-turn reasoning, and (2) clipping-bias correction, which normalizes gradients by downweighting unreliable, highly off-policy samples. Depending on how these components are combined, we obtain three variants: Turn-PPO (turn-level sampling only), **S-PPO** (clipping-bias correction applied to token-level PPO), and **ST-PPO** (turn-level sampling combined with clipping-bias correction). In our experiments, we primarily study ST-PPO and S-PPO, which together demonstrate how the two stabilization mechanisms address complementary sources of instability. Experiments on multi-turn search tasks across general QA, multi-hop QA, and medical multiple-choice QA benchmarks show that ST-PPO and S-PPO consistently prevent the performance collapses observed in large-model training, maintain lower clipping ratios throughout optimization, and achieve higher task performance than standard token-level PPO. These results demonstrate that combining turn-level importance sampling with clipping-bias correction provides a practical and scalable solution for stabilizing multi-turn LLM agent training.

## 1 Introduction

Reinforcement learning (RL) has significantly advanced the reasoning capabilities of large language models (LLMs), enabling strong performance in domains such as mathematical problem solving (Jaech et al., 2024; Liu et al., 2024; Yu et al., 2025) and code generation (El-Kishky et al., 2025; Cui et al., 2025). Beyond these applications, RL has also shown promise in more agentic settings such as tool learning (Qian et al., 2025; Feng et al., 2025), where models learn to invoke external tools (e.g., web search engines), execute actions, and interact with real-world environments. Recent systems such as Deepseek V3 (Liu et al., 2024) and Kimi V2 (Team et al., 2025) have achieved state-of-the-art performance on both mathematical reasoning (e.g., AIME, Math-500) and agentic benchmarks (Jimenez et al., 2023). Despite these successes, the computational demands of multi-turn RL training pose significant practical challenges. These gains rely on numerous interaction samples, which are costly due to the large number of rollouts and multi-turn tool calls required during training. In practice, hardware and memory limits force each batch of collected samples to be split into several mini-batches (Schulman et al., 2017) and updated sequentially. This naturally induces a hybrid update mechanism where later updates become increasingly off-policy (Chen et al., 2023). To maximize sample efficiency under computational constraints, practitioners often adopt off-policy pipelines, and the resulting distribution mismatch is typically corrected using importance sampling (Nachum et al., 2017).

The shift to off-policy methods, while necessary for computational efficiency, introduces high variance and can destabilize training (Munos et al., 2016; Precup et al., 2000). To address this challenge,

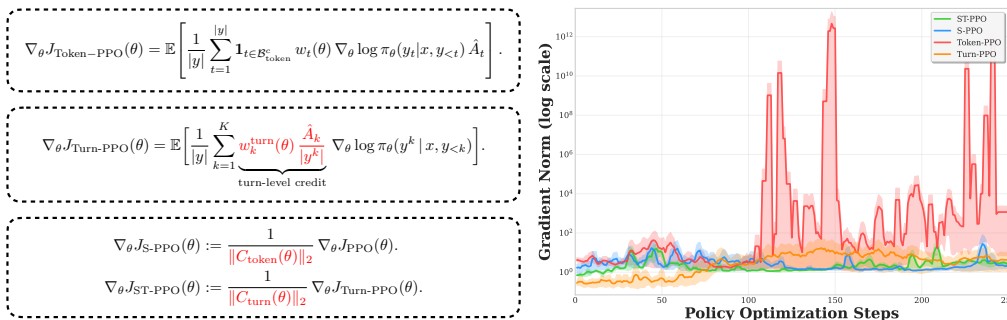

$$\nabla_\theta J_{\text{Token-PPO}}(\theta) = \mathbb{E}\left[\frac{1}{|y|}\sum_{t=1}^{|y|} \mathbf{1}_{t \in \mathcal{B}_{\text{token}}^c}\, w_t(\theta)\, \nabla_\theta \log \pi_\theta(y_t|x, y_{<t})\, \hat{A}_t\right].$$

$$\nabla_\theta J_{\text{Turn-PPO}}(\theta) = \mathbb{E}\left[\frac{1}{|y|}\sum_{k=1}^{K} \underbrace{w_k^{\text{turn}}(\theta)\, \frac{\hat{A}_k}{|y^k|}}_{\text{turn-level credit}}\, \nabla_\theta \log \pi_\theta(y^k \,|\, x, y_{<k})\right].$$

$$\nabla_\theta J_{\text{S-PPO}}(\theta) := \frac{1}{\|C_{\text{token}}(\theta)\|_2}\, \nabla_\theta J_{\text{PPO}}(\theta).$$

$$\nabla_\theta J_{\text{ST-PPO}}(\theta) := \frac{1}{\|C_{\text{turn}}(\theta)\|_2}\, \nabla_\theta J_{\text{Turn-PPO}}(\theta).$$

Figure 1: Illustration of the four PPO variants. Token-level PPO becomes Turn-level PPO by applying turn-level importance sampling (Eq. 4). Further adding the clipping bias to normalize gradients yields S-PPO and ST-PPO (Eq. 8 and Eq. 7). Both variants significantly reduce the probability of extreme gradient spikes, leading to more stable training.

proximal policy optimization (PPO) (Schulman et al., 2017) constrains policy updates with a clipped surrogate objective. By limiting the impact of importance sampling ratios, PPO stabilizes learning even when mini-batch reuse makes later updates effectively off-policy. Building on this principle, recent advances such as TOPR (Roux et al., 2025), LUFFY (Yan et al., 2025), and GSPO (Zheng et al., 2025) refine importance sampling and clipping strategies to further enhance stability and efficiency.

Despite these advances, PPO training on multi-turn tasks still suffers from two core deficiencies: (1) token-level importance sampling misaligns with the natural granularity of multi-turn environments (Zeng et al., 2025), where reasoning unfolds through distinct turn-level stages (e.g., problem analysis, query formulation, information processing), and (2) off-policy updates rely on critic-based advantage estimates that are often unreliable for out-of-distribution tokens (Dorka et al., 2022), leading to high-variance gradients and potential collapse even under clipping.

To address the first deficiency in off-policy RL, we propose **turn-level importance sampling**, as formalized in Lemma 4.1, which establishes the mathematical foundation for turn-level credit assignment. This formulation aligns optimization with the natural boundaries of reasoning and retrieval phases, providing more precise credit assignment while preserving stability. It enables the model to treat different stages of a reasoning trajectory with appropriate emphasis. Such structure-aware optimization proves particularly valuable for search-augmented reasoning, where the quality of intermediate decisions directly impacts final outcomes. The intuition behind turn-level importance sampling lies in balancing granularity and stability: token-level methods are overly noisy, sequence-level methods overly coarse, while turn-level ratios strike the middle ground by capturing sub-goal level contributions without amplifying token-level variance.

To further stabilize off-policy training and address the second deficiency, we employ an **adjustment for clipping-induced bias** as detailed in Lemma 4.2, which decomposes PPO's gradient to isolate the clipping bias term. Retaining token-level information constrains high-variance updates, and turn-level importance sampling guides credit assignment. This hybrid design uses turn-level credit assignment and token-level clipping bias for stabilization, achieving a balance between stability and fidelity and ensuring more conservative and robust policy updates.

**Our key contributions are as follows:**

1. We empirically diagnose why vanilla PPO becomes unstable when applied to multi-turn LLM agent training under off-policy updates. Our observations reveal two root causes unique to the agentic setting: (i) the granularity mismatch between token-level optimization and turn-level interactions, and (ii) the accumulation of variance from unreliable critic estimates on off-policy samples, where state-action pairs are poorly evaluated.

2. We propose a PPO variant based on turn-level importance sampling (Turn-PPO) that better matches the structure of multi-turn reasoning and enables turn-level credit assignment.

3. We propose **stabilized turn-level PPO** (ST-PPO, see Fig. 1, Section 4), a hybrid algorithm that combines turn-level ratios for optimization with a clipping-bias correction to normalize gradient updates. This design reduces highly off-policy updates and leads to more conservative and robust policy learning.

These contributions constitute the first tailored adaptation of PPO for multi-turn LLM agents, integrating theoretical insights with algorithmic design. By addressing both the granularity mismatch and the instability of off-policy updates, our approach provides a principled framework for stable optimization. Empirically, it mitigates the collapse observed in large-model training and consistently improves task performance, offering a practical and scalable solution for reinforcement learning with multi-turn LLM agents.

## 2 RELATED WORKS

### 2.1 REINFORCEMENT LEARNING WITH LLM AGENTS

Recent advances in reinforcement learning for large language models (LLMs) have primarily progressed along two main directions: (1) feedback-driven alignment and policy optimization, and (2) agentic tool use with long-horizon reasoning.

On the alignment side, RLHF (Ouyang et al., 2022) translates human or rule-based preferences into reward signals and optimizes policies using policy gradient methods, as exemplified by InstructGPT. Subsequent approaches such as direct preference optimization (DPO) (Rafailov et al., 2023) avoid explicit reward modeling, while other RL algorithms—including PPO (Schulman et al., 2017), GRPO (Shao et al., 2024), and RLOO (Ahmadian et al., 2024)—have been adapted to large models to balance stability and efficiency.

On the agentic side, methods such as ReAct (Yao et al., 2023), Give (He et al., 2024; 2025) integrate reasoning and acting through interleaved thought–action steps, while Reflexion (Shinn et al., 2023) leverages self-reflection and memory to enhance multi-turn decision making. Toolformer (Schick et al., 2023) demonstrates self-supervised learning of tool usage, and WebGPT (Nakano et al., 2021) introduces browser-based agents trained with human feedback. More recent systems, such as Search-R1 (Jin et al., 2025b), WebAgent-R1 (Wei et al., 2025), and WEBRL (Qi et al., 2024), extend this line of research to end-to-end reinforcement learning for web-based agents, where models are trained to perform retrieval, reasoning, and action within real-world interactive environments.

### 2.2 OFF POLICY REINFORCEMENT LEARNING WITH LLM AGENTS

Off-policy reinforcement learning methods differ fundamentally from on-policy approaches by allowing the agent to learn from data generated by a policy different from the one currently being improved. This characteristic enables greater flexibility and data efficiency, as off-policy algorithms can reuse past experiences or observations collected under different policies. This property is particularly important in scenarios where data collection is costly or limited.

Recent work has extended these ideas to policy gradient methods tailored for off-policy learning in complex environments. The tapered off-policy REINFORCE (TOPR) algorithm (Roux et al., 2025) builds on the classic REINFORCE method (Williams, 1992) by introducing a tapered importance sampling scheme for the policy gradient. This tapering reduces the variance and instability that commonly arise in naive off-policy gradient methods, as it asymmetrically adjusts the importance weights to better balance learning from positive and negative examples. However, because TOPR relies heavily on trajectory-level rewards, it is not well-suited for scenarios that require more fine-grained credit assignment.

Another research line explores off-policy extensions of GRPO. ARPO (Lu et al., 2025) incorporates a replay buffer into GRPO's sampling process, mixing on-policy and replayed samples to alleviate the zero-advantage issue, while RePO (Li et al., 2025a) reuses past outputs as off-policy samples with tailored replay strategies (e.g., recency-based, reward-oriented) to improve data efficiency and stability. Although these methods enhance sample utilization, they still require substantial training time, and their trajectory-level credit assignment often leads to slower convergence on multi-turn tasks.

### 2.3 REINFORCEMENT LEARNING IN MULTI-TURN TASKS

Recent advances in LLMs have increasingly leveraged reinforcement learning to enhance multi-turn reasoning and tool use capabilities (Chen et al., 2025; Cheng et al., 2025; Li et al., 2025c). The work on turn-level credit assignment by Zeng et al. (2025) introduces a novel RL framework that models multi-turn interactive tasks as Markov decision processes, enabling fine-grained credit allocation to individual reasoning and tool-use steps within a trajectory. Complementing this, the Search-R1 framework proposed by Jin et al. (2025b;a) similarly harnesses RL to train LLMs to reason

about when and how to use external search engines interactively. By framing the reasoning and search process as an end-to-end decision-making problem, Search-R1 encourages LLMs to generate informative queries and iteratively refine answers based on real-time feedback from the search tool. This work emphasizes the importance of multi-turn interactions to bridge the gap between internal textual reasoning and external information retrieval, demonstrating improved reasoning robustness and answer quality over traditional single-turn or passive retrieval approaches. Despite these advances, there remains a lack of algorithms specifically tailored to stabilize PPO in multi-turn LLM off-policy training. Our work addresses this gap by introducing a customized PPO variant that explicitly leverages turn-level structure, leading to more stable and scalable training.

## 3 PRELIMINARIES

**Notation.** An autoregressive language model parameterized by $\theta$ is defined as a policy $\pi_\theta$. We use $x$ to denote a query and $\mathcal{D}$ as the query set. Given a response $y$ to a query $x$, its likelihood under the policy $\pi_\theta$ is denoted as $\pi_\theta(y \mid x) = \prod_{t=1}^{|y|} \pi_\theta(y_t \mid x, y_{<t})$, where $|y|$ denotes the number of tokens in $y$. A query-response pair $(x, y)$ can be scored by a verifier $r$, resulting in a reward $r(x, y) \in [0, 1]$.

**Turn-level MDP.** We model multi-turn interactions as a Markov decision process $\mathcal{M} = (\mathcal{S}, \mathcal{A}, \mathcal{P}, \mathcal{R}, \gamma)$ defined at the granularity of turns. A turn is denoted by $y^k \coloneqq (y_{t_k^{\text{start}}}, \ldots, y_{t_k^{\text{end}}})$, and a full trajectory can be written as $y \coloneqq (y^1; \ldots; y^n)$ for $n$ turns, where ';' denotes token concatenation. Each state $s_k \in \mathcal{S}$ corresponds to the dialogue context at the beginning of turn $k$, which we define as $s_k \coloneqq (x, y^1, y^2, \ldots, y^{k-1})$. The transition $\mathcal{P}(s_{k+1} \mid s_k, a_k)$ updates the context with environment feedback (e.g., results from a tool call), and the reward $\mathcal{R}(s_k, a_k)$ evaluates the quality of the turn. The discount factor $\gamma \in (0, 1]$ balances present and future rewards. Unlike token-level MDPs, the unit of interaction here is the *turn*, with boundaries $[t_k^{\text{start}}, t_k^{\text{end}}]$, aligning credit assignment with the natural structure of reasoning and retrieval phases.

**Proximal Policy Optimization (PPO).** Using samples generated from the old policy $\pi_{\theta_{\text{old}}}$, PPO constrains the policy update within a proximal region of the old policy through the clipping mechanism. This allows for multiple gradient updates to the policy using the same batch of samples. Specifically, PPO employs the following objective for policy optimization:

$$\mathcal{J}_{\text{PPO}}(\theta) = \mathbb{E}_{x \sim \mathcal{D}, \, y \sim \pi_{\theta_{\text{old}}}(\cdot|x)} \left[ \frac{1}{|y|} \sum_{t=1}^{|y|} \min \left( w_t(\theta) \hat{A}_t, \, \text{clip}\left(w_t(\theta), 1 - \epsilon, 1 + \epsilon\right) \hat{A}_t \right) \right], \quad (1)$$

where the importance ratio of the token $y_t$ is defined as

$$w_t(\theta) = \frac{\pi_\theta(y_t \mid x, y_{<t})}{\pi_{\theta_{\text{old}}}(y_t \mid x, y_{<t})}, \quad (2)$$

where $\hat{A}_t$ denotes the token-level advantage estimated by GAE based on the critic model $\hat{V}$. As discussed in related work, PPO has been widely used in many agentic tasks and performs well in single-turn settings. However, its performance in multi-turn settings is often unstable and prone to collapse; for example, Yuan et al. (2025) show that PPO can fail in long reasoning tasks. In multi-turn settings, reward signals are typically provided only at the end of a long interaction. This delayed feedback leaves value predictions at intermediate states poorly constrained, and the resulting errors propagate backward through temporal-difference updates, compounding over many steps (Arjona-Medina et al., 2019). Such delayed and sparse rewards substantially increase bias and variance in value estimation. In addition, under off-policy updates the critic must evaluate state–action pairs that may be rarely or never visited by the current policy, further exacerbating estimation errors (Munos et al., 2016; Dorka et al., 2022). To summarize, PPO deficiencies in multi-turn environments can be summarized as follows:

1. PPO trains at the token level, which does not match the turn-level structure of multi-turn reasoning. Under off-policy updates, this mismatch becomes more pronounced and often increases the variance of the importance weights.

2. Off-policy samples in multi-turn tasks frequently contain state-action pairs that the critic has not learned to evaluate well. This leads to unreliable advantage estimates, highly off-policy gradients, and in practice, unstable training even with clipping.

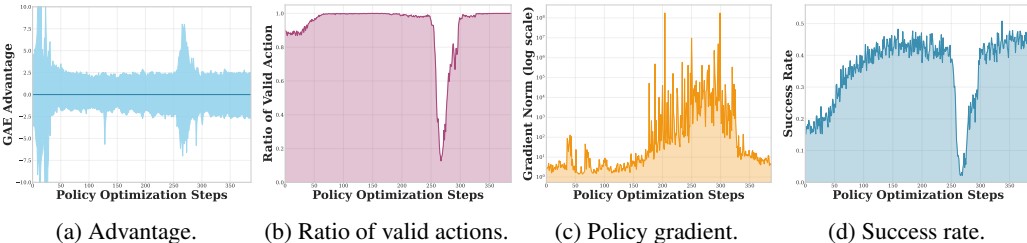

| (a) Advantage. | (b) Ratio of valid actions. | (c) Policy gradient. | (d) Success rate. |

Figure 2: Observations from a failed run with Qwen2.5-7B base model when running token-level PPO. From left to right, we show the estimated advantage, the ratio of valid actions (whether the tool is successfully called), the L2 norm of the policy gradient, and the success rate of the search task. Each metric is recorded for every training batch.

To further probe instability, we examine a failed run of Qwen2.5-7B under token-level PPO (Fig. 2) in multi-turn search tasks. The results highlight two critical issues: (1) around steps 250–300, the critic assigns highly variable advantages while the ratio of valid actions collapses, a typical signature of reward hacking; and (2) these unreliable samples trigger an explosion in the policy gradient norm, which rapidly degrades performance and causes a collapse in task success rate. This evidence confirms that critic estimation errors and variance accumulation can destabilize off-policy updates in multi-turn settings, motivating the need for stabilization mechanisms beyond turn-level scaling.

## 4 PROPOSED ALGORITHM

In this section, we present our approach for stabilizing PPO in multi-turn LLM training. Based on the deficiencies identified in Section 3, we first introduce a turn-level variant of PPO that better matches the granularity of multi-turn reasoning and interaction (Section 4.1). We then propose a stabilization mechanism that leverages clipping bias to adaptively downweight highly off-policy updates (Section 4.2).

### 4.1 TURN-LEVEL PPO FORMULATION

We begin by examining how these deficiencies limit the performance of standard PPO and then introduce a turn-level variant designed to address them For $K$ total turns, with $[t_k^{\text{start}}, t_k^{\text{end}}]$ denoting the token positional boundaries for turn $k \in \{1, \ldots, K\}$, we propose the turn-level PPO formulation:

$$\mathcal{J}_{\text{Turn-PPO}}(\theta) = \mathbb{E}_{x \sim \mathcal{D}, y \sim \pi_{\theta_{\text{old}}}(\cdot|x)} \left[ \frac{1}{|y|} \sum_{k=1}^{K} \sum_{t=t_k^{\text{start}}}^{t_k^{\text{end}}} \min\left\{ w_k^{\text{turn}}(\theta)\hat{A}_t, \text{clip}(w_k^{\text{turn}}(\theta), 1-\epsilon, 1+\epsilon)\hat{A}_t \right\} \right],$$
$$(3)$$

where $w_k^{\text{turn}}$ is a turn-level importance sampling weight. Inspired by the sequence-level weight used by Zheng et al. (2025), we define a turn-level variant as:

$$w_k^{\text{turn}}(\theta) := \left( \frac{\pi_\theta(y^k \mid x, y^{<k})}{\pi_{\theta_{\text{old}}}(y^k \mid x, y^{<k})} \right)^{\frac{1}{|y^k|}} = \exp\left( \frac{1}{|y^k|} \sum_{t=t_k^{\text{start}}}^{t_k^{\text{end}}} \log \frac{\pi_\theta(y_t \mid x, y_{<t})}{\pi_{\theta_{\text{old}}}(y_t \mid x, y_{<t})} \right), \quad (4)$$

where the ratio is scaled by the length of the $k^{\text{th}}$ turn, $|y^k|$, to stabilize the objective across turns of varying lengths. At first glance, this modification appears minimal, as it only alters the computation of importance weights relative to the original PPO formulation in Eq. 1. However, as we formalize below, it fundamentally changes the structure of credit assignment. To prepare for this analysis, we define the events under which clipping is inactive for the turn-level objective in Eq. 3:

$$\mathcal{B}_{\text{turn}}^+ := \{ (k,t) : \hat{A}_t \geq 0, \ w_k^{\text{turn}}(\theta) \leq 1+\epsilon \}, \qquad \mathcal{B}_{\text{turn}}^- := \{ (k,t) : \hat{A}_t < 0, \ w_k^{\text{turn}}(\theta) \geq 1-\epsilon \}.$$

Let $\mathcal{B}_{\text{turn}} := \mathcal{B}_{\text{turn}}^+ \cup \mathcal{B}_{\text{turn}}^-$. When $\mathcal{B}_{\text{turn}}$ holds, the PPO objective reduces to the unclipped form $w_t \hat{A}_t$, so the update is fully determined by the critic's advantage estimates. In contrast, when $\mathcal{B}_{\text{turn}}^c$ occurs (clipping is active), it indicates a mismatch between the reference and current policies, and gradients of the corresponding tokens are set to zero.

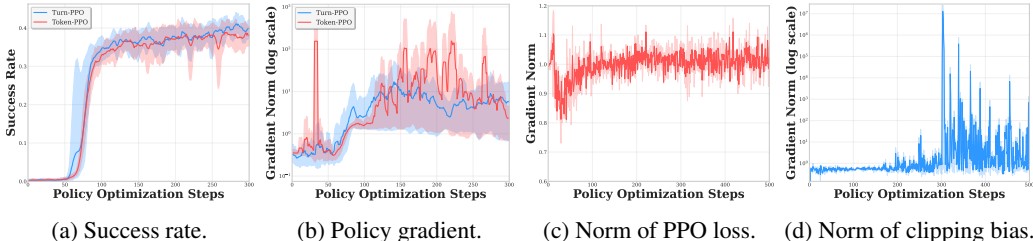

(a) Success rate.      (b) Policy gradient.      (c) Norm of PPO loss.      (d) Norm of clipping bias.

Figure 3: Comparison of token-level versus turn-level PPO training on Qwen2.5-1.5B for the search task (results averaged over 5 runs). (a) Success rates demonstrate that turn-level PPO outperforms compared to token-level PPO. (b) L2 norms of policy gradients show that turn-level PPO exhibits greater training stability. (c–d) Additional diagnostic metrics for token-level PPO: (c) the L2 norm of the PPO loss function remains stable throughout training due to gradient clipping, while (d) the L2 norm of the clipping bias term grows quickly, validating Lemma 4.2.

**Lemma 4.1.** *The gradient of the objective function in Eq. 3 is given by*

$$\nabla_\theta \mathcal{J}_{\text{Turn-PPO}}(\theta) = \mathbb{E}\left[\frac{1}{|y|}\sum_{k=1}^{K}\underbrace{w_k^{turn}(\theta)\frac{\hat{A}^k}{|y^k|}}_{\textit{Turn-Level Credit}}\nabla_\theta\log\pi_\theta(y^k|x,y^{<k})\right], \tag{5}$$

*where* $|y^k| = t_k^{end} - t_k^{start} + 1$ *and* $\hat{A}^k := \sum_{t=t_k^{start}}^{t_k^{end}}\mathbb{1}_{\{(k,t)\in\mathcal{B}_{turn}\}}\hat{A}_t$.

*Proof.* We refer to Appendix A.2 for the proof of Lemma 4.1. $\qquad\square$

**Remark.** Lemma 4.1 highlights the effect of modifying the importance sampling ratio from the token level to the turn level. As shown in Eq. 5, all tokens within a turn share the same aggregated advantage $\hat{A}^k$, even though both the advantages and the clipping mechanism are still computed token-wise. The turn-level structure arises because tokens in the same turn use a common geometric mean importance ratio, and the final gradient is obtained by aggregating the unclipped token-level contributions within $\mathcal{B}_{\text{turn}}$. Consistent with our following experiments and prior studies (Zhou et al., 2024; Zeng et al., 2025), incorporating turn-level credit assignment generally leads to improved performance on multi-turn tasks.

**Validation.** We first evaluate the performance of token-level PPO in Eq. 1 and turn-level PPO in Eq. 3 on the search task using the Qwen2.5-1.5B base model. Fig. 3 reports the average results over five runs. As shown in Fig. 3a, turn-level PPO achieves a higher performance (success rate) than token-level PPO, confirming that eliminating the granularity mismatch improves task performance. Moreover, Fig. 3b shows that the average policy gradient norm of turn-level PPO is consistently lower, suggesting that turn-level scaling helps stabilize training.

These results demonstrate that turn-level PPO alleviates part of the instability. However, when tested with larger models such as Qwen2.5-7B, we observe that both token-level and turn-level PPO still suffer from collapse. This observation motivates the development of additional stabilization mechanisms, which we introduce in the next section.

### 4.2 STABILIZING OFF-POLICY TRAINING VIA CLIPPING-BIAS CORRECTION

While turn-level importance sampling improves stability, it does not fully prevent collapse, as highly off-policy or unreliable samples can still destabilize updates. To address this, we decompose the gradient of the token-level PPO objective by first defining the events under which clipping is inactive. Specifically, we begin by defining the events under which clipping is inactive:

$$\mathcal{B}_{\text{token}}^+ := \{\, t : \hat{A}_t \geq 0,\ w_t(\theta) \leq 1+\epsilon \,\}, \qquad \mathcal{B}_{\text{token}}^- := \{\, t : \hat{A}_t < 0,\ w_t(\theta) \geq 1-\epsilon \,\}.$$

Let $\mathcal{B}_{\text{token}} := \mathcal{B}_{\text{token}}^+ \cup \mathcal{B}_{\text{token}}^-$. With this notation in place, we can formally decompose the PPO gradient as follows:

---

**Algorithm 1:** *Multi-turn PPO with Token- and Turn-Level Importance Sampling (ST-PPO)*

---

**Input:** Initialize policy $\pi_0$, step size $\eta$, dataset $\mathcal{D}$, and maximum generation length $L$.

**for** iteration $k = 0, 1, \ldots, K-1$ **do**

    **Data Sampling I:** Sample a batch of queries $x \sim \mathcal{D}$ and generate agent trajectories $\tau := y_{0:L} \sim \pi_k$.

    **Data Sampling II:** For each trajectory, split it into turn-level state–action pairs $(s_0, a_0), (s_1, a_1), \ldots$ (e.g., using a loss mask or `<eot>` tokens).

    **Gradient Estimation:** Compute the Turn-PPO gradient as in Eq. 3.

    **Surrogate Gradient:** Use token-level importance sampling to compute the clipping-error norm and form the surrogate gradient in Eq. 7.

    **Policy Improvement:** Update the policy using the surrogate gradient in Eq. 7.

    **Policy Evaluation:** Update the critic using the temporal difference (TD) error.

**end for**

---

**Lemma 4.2.** *The PPO gradient can be decomposed as*

$$\frac{\partial \mathcal{J}_{\text{PPO}}(\theta)}{\partial \log \pi_\theta} = \underbrace{\mathbb{E}\left[\frac{1}{|y|}\sum_{t=1}^{|y|} w_t A_t^\pi\right]}_{\textit{Off-Policy term}} + \underbrace{\mathbb{E}\left[\frac{1}{|y|}\sum_{t=1}^{|y|} w_t \left(\hat{A}_t - A_t^\pi\right)\right]}_{\textit{Advantage Estimation Error Term}} - \underbrace{\mathbb{E}\left[\frac{1}{|y|}\sum_{t=1}^{|y|} \mathbb{1}_{t\in\mathcal{B}_{token}^c}\, w_t \hat{A}_t\right]}_{C(\theta): \textit{ Clipping Bias Term}},$$

$$(6)$$

*where $\hat{A}_t$ denotes the advantage estimate by the critic using the GAE method, while $A^\pi$ denotes the advantage estimated by the ground-truth value function $V^\pi$.*

*Proof.* We refer to Appendix A.3 for the proof of Lemma 4.2. $\qquad\square$

From the gradient decomposition, we see that the PPO gradient consists of three terms. The first is the *off-policy term*, corresponding to the gradient of the vanilla policy gradient algorithm without clipping, where all tokens contribute equally to the update. The second is the *advantage estimation error term*, which captures the discrepancy between the critic's estimated advantages and the ground-truth ones, thereby introducing both variance and systematic bias. The third is the *clipping bias term*, arising directly from the indicator $\mathbb{1}_{t\in\mathcal{B}_{token}^c}$ in the clipped objective: while clipping stabilizes training by suppressing large updates, it also discards certain token contributions and hence introduces bias.

We call the third term in Eq. 6 the clipping bias term. To examine its behavior, we conduct an experiment with token-level PPO on the Qwen2.5-7B base model and track the L2 norm of this term across training iterations. For comparison, we also report the L2 norm of PPO's standard loss (Fig. 3c) and the clipping bias norm (Fig. 3d). As expected, the standard PPO loss remains relatively stable across iterations since clipping suppresses extreme updates. In contrast, the clipping bias grows quickly and exhibits large oscillations, indicating that a subset of generations contain outlier tokens with extreme values, rendering these samples unreliable even after clipping.

By focusing updates on more stable, low-error samples, the optimization becomes conservative and avoids destabilizing spikes. Motivated by this observation, we propose a surrogate gradient estimator that reweights each sampled generation according to its clipping error, thereby ensuring stable policy updates. From a scaling perspective, turn-level importance sampling treats each turn as a whole, with a weight given by the turn-level importance ratio multiplied by the cumulative unclipped advantage and normalized by the turn length. To further enhance stability, we incorporate an additional scaling factor derived from the clipping bias term, which downweights highly off-policy or risk-prone samples. This leads to a surrogate objective with distinct clipping-bias corrections for token-level and turn-level PPO, denoted $C_{\text{token}}(\theta)$ and $C_{\text{turn}}(\theta)$, respectively:

$$C_{\text{token}}(\theta) := \mathbb{E}\left[\frac{1}{|y|}\sum_{t=1}^{|y|} \mathbb{1}_{t\in\mathcal{B}_{token}^c}\, w_t \hat{A}_t\right], \quad C_{\text{turn}}(\theta) := \mathbb{E}\left[\frac{1}{|y|}\sum_{t=1}^{|y|} \mathbb{1}_{t\in\mathcal{B}_{turn}^c}\, w_t^{\text{turn}} \hat{A}_t\right].$$

Based on these definitions, the surrogate gradients are given by

$$\nabla_\theta \mathcal{J}_{\text{ST-PPO}}(\theta) := \frac{1}{\|C_{\text{turn}}(\theta)\|_2} \nabla_\theta \mathcal{J}_{\text{Turn-PPO}}(\theta), \tag{7}$$

$$\nabla_\theta \mathcal{J}_{\text{S-PPO}}(\theta) := \frac{1}{\|C_{\text{token}}(\theta)\|_2} \nabla_\theta \mathcal{J}_{\text{PPO}}(\theta). \tag{8}$$

Fig. 1 illustrates the relations among the four PPO variants we study. Together with S-PPO (clipping-bias correction at the token level), these variants complete the set of four algorithms: Token-PPO, Turn-PPO, S-PPO, and ST-PPO. Since Turn-PPO mainly serves as an intermediate variant, our experiments focus on Token-PPO, S-PPO, and ST-PPO. The overall pipeline is summarized in Algorithm 1.

Notably, this clipping-bias correction can also be incorporated into the GRPO framework, yielding a stabilized variant that we refer to as **S-GRPO**. Recall that GRPO replaces the token-level PPO objective with a group-level surrogate based on trajectory-level rewards, where the importance ratio is computed between two complete responses rather than individual tokens. Formally, GRPO optimizes an objective of the form

$$J_{\text{GRPO}}(\theta) = \mathbb{E}\big[\, w_t(\theta)\hat{A} \,\big], \tag{9}$$

where $w_t(\theta)$ is the token level importance ratio and $\hat{A}_{\text{grp}}$ denotes the group advantage. Although GRPO avoids token-level clipping, it still relies on importance sampling under off-policy data and therefore inherits the same vulnerability identified in Lemma 4.2: a small number of highly off-policy samples can dominate the gradient and destabilize training.

To apply our stabilization mechanism, we compute the clipping bias using token-level importance ratios and group advantages. Specifically, define

$$C_{\text{GRPO}}(\theta) := \mathbb{E}\Big[\, \mathbb{1}_{t \in \mathcal{B}^c_{\text{token}}} \, w_t(\theta) \, \hat{A}_t \,\Big]. \tag{10}$$

which measures the contribution of samples whose group-level ratios fall outside the trust region. The stabilized GRPO update is then defined analogously to S-PPO:

$$\nabla_\theta J_{\text{S-GRPO}}(\theta) := \frac{1}{\big\|C_{\text{GRPO}}(\theta)\big\|_2} \, \nabla_\theta J_{\text{GRPO}}(\theta).$$

This yields a lightweight but effective variant: S-GRPO rescales each update by the inverse clipping-bias norm, thereby downweighting trajectories exhibiting extreme off-policy behavior. We do not apply turn-level importance sampling here because GRPO's credit assignment is already fixed at the trajectory level, leaving no natural turn-level structure to exploit; In practice, this removes a major source of instability in GRPO, especially under off-policy pipelines.

**Turn-Level Credit Assignment.** From Lemma 4.1, after revising the importance sampling, our algorithm is able to assign turn-level credit to each interaction, independent of the specific choice of advantage estimator (e.g., GAE or Monte Carlo). This removes the mismatch between the level of interaction (turns) and the level at which credit is assigned.

**Optimality.** In Eqs. 7 and 8, we stabilize the gradient by normalizing it with the clipping bias term. For each sample, both the original turn-level PPO and ST-PPO update along gradient directions that are aligned, differing only by normalization. As a result, ST-PPO preserves the same optimality conditions as Turn-PPO while providing more stable policy improvement.

**Generality.** The formulation is flexible enough to recover existing LLM training methods (e.g., GRPO, RLOO) through specific choices of advantage design. This generality allows algorithms developed for advantage estimation to be directly applied as special cases within our framework.

## 5 EXPERIMENTS

We conduct comprehensive numerical evaluations of our proposed ST-PPO algorithm (Algorithm 1) and compare its performance against the state-of-the-art baseline, Search-R1 Jin et al. (2025b). Our experimental results demonstrate two key advantages of the proposed approach: (1) the combination

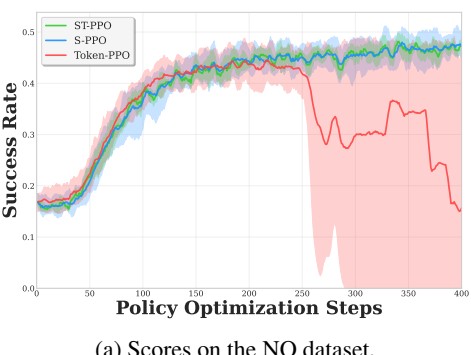 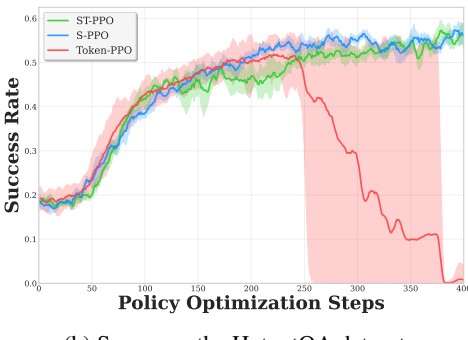

(a) Scores on the NQ dataset.      (b) Scores on the HotpotQA dataset.

Figure 4: Experimental results of Qwen-2.5-7B policy models, with the value model also trained from Qwen-2.5-7B. Results are averaged over three trials. We report the average success rate on the NQ and HotpotQA datasets.

of turn-level importance sampling and clipping-bias correction significantly enhances training stability compared to standard PPO, preventing the performance collapses commonly observed in multi-turn agent training, and (2) ST-PPO and S-PPO maintain consistently lower clipping ratios throughout training, indicating more reliable gradient updates and improved sample efficiency.

**Experimental Setup.** Following the experimental setup of Search-R1 Jin et al. (2025b), we conduct our experiments using the Qwen-2.5-7B (Base) model (Bai et al., 2023) as the base policy. For retrieval, we employ the 2018 Wikipedia dump (Karpukhin et al., 2020) as our knowledge source and the E5 embeddings (Wang et al., 2022) as the retriever. To ensure fair comparison with existing methods, we follow Lin et al. (2023) and consistently retrieve three passages across all retrieval-based approaches. We evaluate ST-PPO on two benchmark datasets: (1) Natural Questions (NQ)(Kwiatkowski et al., 2019) and (2) HotpotQA(Ho et al., 2020). These datasets present diverse multi-hop reasoning and search challenges that require effective coordination between information retrieval and reasoning steps, making them ideal testbeds for evaluating the stability and effectiveness of our turn-level optimization approach. For complete implementation details, refer to Appendix A.4.

**Evaluation.** For evaluation, we evaluate on the test sets of the NQ and HotpotQA datasets to assess model performance on each domain. Exact Match (EM) is used as the primary evaluation metric, which measures the percentage of predictions that exactly match the ground-truth answers after standard normalization. EM provides a strict assessment of whether our multi-turn approach successfully retrieves and synthesizes correct information through the search and reasoning process.

**Training Stability and Performance.** Fig. 4 compares the training dynamics of Token-level PPO, ST-PPO, and S-PPO on the NQ and HotpotQA datasets. All methods demonstrate rapid initial improvement, confirming that PPO can identify effective policies early in training. However, their long-term behaviors diverge dramatically. Token-level PPO exhibits marked instability: after reaching peak performance, we observe sharp performance collapses that render the learned policy significantly worse than earlier checkpoints. This degradation pattern aligns with our theoretical analysis, where we identified that critic estimation errors on tokens with high importance weights can destabilize gradient updates. Such instability necessitates careful early stopping to preserve the best-performing checkpoint, complicating the training process. In contrast, ST-PPO and S-PPO maintain consistent upward progress throughout the entire optimization horizon without performance collapses.

Fig. 6a further shows that both ST-PPO and S-PPO achieve significantly lower clipping ratios than PPO, while Fig. 6b demonstrates consistently smaller KL divergence during optimization. These results confirm that our methods reduce the magnitude of unstable updates and improve training safety. In Appendix A.5, we conduct an experiment under a more off-policy setting, and the results show that our method remains more stable than the baseline algorithm.

This stability stems from two key mechanisms: (1) the turn-level importance sampling aligns credit assignment with the natural structure of reasoning-search interactions, reducing the granularity mismatch that contributes to instability, and (2) the clipping-bias correction adaptively scales gradient updates based on sample reliability, reducing the influence of tokens with high variance or poor critic estimates. As a result, ST-PPO and S-PPO achieve both competitive peak performance and robust convergence without requiring early stopping or careful checkpoint selection.

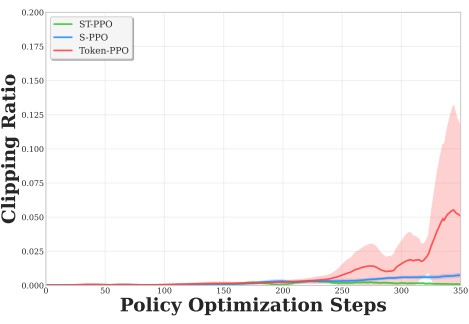
(a) Clipping Ratio on the HotpotQA dataset.

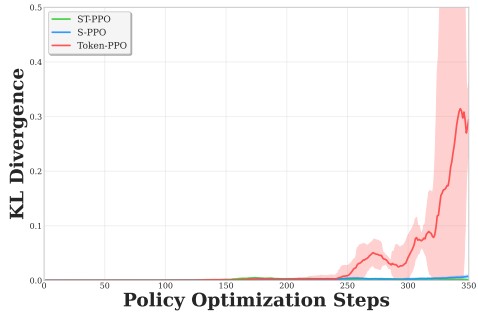
(b) KL divergence on the HotpotQA dataset.

Figure 5: We also report (a) the clipping ratio and (b) the KL divergence during policy optimization. Both ST-PPO and S-PPO achieve lower clipping ratios and KL divergence compared to vanilla PPO, indicating more stable training dynamics.

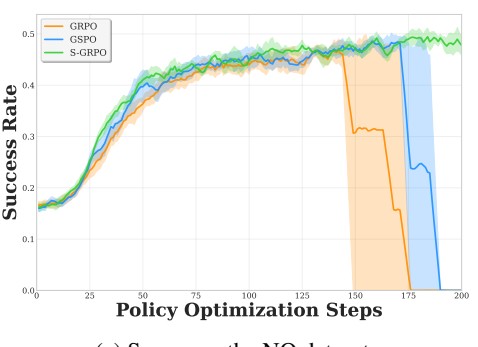
(a) Scores on the NQ dataset.

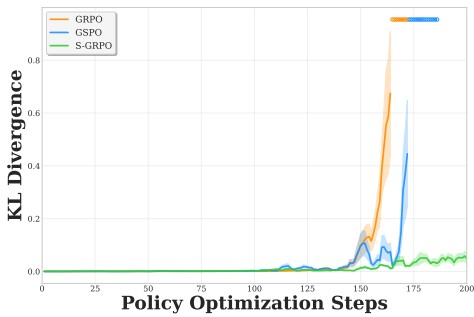
(b) KL divergence on the NQ dataset.

Figure 6: Comparison of GRPO, GSPO, and S-GRPO on the NQ dataset. (a) Success rates show that GRPO and GSPO collapse during training, whereas S-GRPO remains stable. (b) KL divergence exhibits large spikes for GRPO and GSPO, while S-GRPO stays near zero. Circles mark steps where the gradient becomes NaN.

To further validate the effectiveness of our GRPO variant, we also conduct a comparison among GRPO, GSPO, and our proposed S-GRPO on the NQ dataset. The results in Figure 6 show that S-GRPO exhibits substantially improved training stability compared to both GRPO and GSPO, with markedly smoother loss-norm curves throughout optimization. This confirms that clipping-bias normalization serves as an effective stabilization mechanism even under trajectory-level objectives such as GRPO.

## 6 CONCLUSION

In this work, we analyzed why PPO becomes unstable in multi-turn LLM agent training and identified two main causes: the mismatch between token-level optimization and turn-level credit assignment, and the high variance from off-policy tokens. Through a theoretical decomposition of PPO's gradient updates, we showed how these factors lead to unstable convergence and occasional collapse. To address this, we proposed ST-PPO, which integrates token- and turn-level importance sampling with clipping-bias correction to yield more conservative and stable updates. Empirically, ST-PPO produced consistently smooth training curves, whereas PPO often required early stopping. These results highlight the value of matching optimization granularity with task structure and point toward safer reinforcement learning methods for multi-turn LLM agents.

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

# A  APPENDIX

## A.1  LLM USAGE

In this work, LLM were used solely for polishing the writing. No part of the technical content, experimental design, or analysis relied on LLMs. The responsibility for the correctness and originality of the ideas, methods, and results remains entirely with the authors.

## A.2  PROOF OF LEMMA 4.1

The proof establishes turn-level credit assignment by first reformulating the PPO objective with indicator functions, then computing how the gradient with respect to a single token depends only on its containing turn due to locality. The key insight is that the turn-level importance weight derivative yields a scaling factor of $\frac{w_k^{\text{turn}}(\theta)}{|y^k|}$, where the $\frac{1}{|y^k|}$ term arises from the geometric mean structure of the turn-level importance ratio. Finally, aggregating over all tokens via the multivariable chain rule produces the claimed gradient form where each turn receives credit proportional to $w_k^{\text{turn}}(\theta)\frac{\hat{A}^k}{|y^k|}$, combining importance weighting, accumulated advantage, and length normalization.

We now proceed with the lemma proof. We derive the gradient of the Turn-PPO objective $\mathcal{J}_{\text{Turn}-\text{PPO}}(\theta)$ defined in Eq. 3. Beginning with the objective in expanded form:

$$\mathcal{J}_{\text{Turn}-\text{PPO}}(\theta)$$

$$= \mathbb{E}_{x\sim\mathcal{D},\, y\sim\pi_{\theta_{\text{old}}}(\cdot|x)}\left[\frac{1}{|y|}\sum_{k=1}^{K}\sum_{t=t_k^{\text{start}}}^{t_k^{\text{end}}}\min\left\{w_k^{\text{turn}}(\theta)\hat{A}_t, \text{clip}(w_k^{\text{turn}}(\theta), 1-\epsilon, 1+\epsilon)\hat{A}_t\right\}\right]$$

$$= \mathbb{E}_{x\sim\mathcal{D},\, y\sim\pi_{\theta_{\text{old}}}(\cdot|x)}\left[\frac{1}{|y|}\sum_{k=1}^{K}\sum_{t=t_k^{\text{start}}}^{t_k^{\text{end}}}\left(\mathbb{1}_{(k,t)\in\mathcal{B}_{\text{turn}}}w_k^{\text{turn}}(\theta)\hat{A}_t + \mathbb{1}_{(k,t)\in B_{\text{turn}}^c}\text{clip}(w_k^{\text{turn}}(\theta), 1-\epsilon, 1+\epsilon)\hat{A}_t\right)\right],$$

$$\tag{11}$$

where[1] Eq. 11 follows from decomposing the min function using indicator functions. The indicator function $\mathbb{1}_{(k,t)\in\mathcal{B}_{\text{turn}}} = 1$, if the token index $t$ within turn $k$ belongs to the set $\mathcal{B}_{\text{turn}}$ and 0 otherwise. The set $\mathcal{B}_{\text{turn}} = \mathcal{B}_{\text{turn}}^{+} \cup \mathcal{B}_{\text{turn}}^{-}$ represents the indices of tokens within turns where clipping is inactive. Together, the set $\mathcal{B}_{\text{turn}}$ and its complement $\mathcal{B}^c$ partition the space of token indices across all turns, satisfying $\mathcal{B}_{\text{turn}} \cup \mathcal{B}_{\text{turn}}^c = \{1, 2, \ldots, |y|\}$ and $\mathcal{B}_{\text{turn}} \cap \mathcal{B}_{\text{turn}}^c = \emptyset$. For token indices $(k,t) \in \mathcal{B}_{\text{turn}}$, the min operation selects the unclipped term $w_k^{\text{turn}}(\theta)\hat{A}_t$; otherwise, for $t \in \mathcal{U}^c$, it selects the clipped term $\text{clip}(w_k^{\text{turn}}(\theta), 1-\epsilon, 1+\epsilon)\hat{A}_t$.

Taking the gradient with respect to $\log\pi_\theta(y_{t'}|x, y_{<t'})$ where $t' \in [t_{k'}^{\text{start}}, t_{k'}^{\text{end}}]$ (i.e., token $t'$ belongs to turn $k'$), we get

$$\frac{\partial\mathcal{J}_{\text{Turn}-\text{PPO}}(\theta)}{\partial\log\pi_\theta(y_{t'}|x, y_{<t'})} = \mathbb{E}\left[\frac{1}{|y|}\sum_{k=1}^{K}\sum_{t=t_k^{\text{start}}}^{t_k^{\text{end}}}\mathbb{1}_{(k,t)\in\mathcal{B}_{\text{turn}}}\frac{\partial w_k^{\text{turn}}(\theta)}{\partial\log\pi_\theta(y_{t'}|x, y_{<t'})}\hat{A}_t\right]\tag{12}$$

$$= \mathbb{E}\left[\frac{1}{|y|}\sum_{t=t_{k'}^{\text{start}}}^{t_{k'}^{\text{end}}}\mathbb{1}_{(k,t)\in\mathcal{B}_{\text{turn}}}\frac{\partial w_{k'}^{\text{turn}}(\theta)}{\partial\log\pi_\theta(y_{t'}|x, y_{<t'})}\hat{A}_t\right],\tag{13}$$

where Eq. 12 follows since clipped terms have zero gradients by definition of the clipping function; and Eq. 13 follows since $w_k^{\text{turn}}(\theta)$ depends only on tokens within turn $k$. Since token $t'$ belongs to turn $k'$, we have $\frac{\partial w_k^{\text{turn}}(\theta)}{\partial\log\pi_\theta(y_{t'}|x,y_{<t'})} = 0$ for all $k \neq k'$ due to disjoint turn boundaries.

---

[1]For notational brevity, we drop the expectation subscript in what follows, with all expectations taken over the same joint distribution where $x \sim \mathcal{D}$ and $y \sim \pi_{\theta_{\text{old}}}(\cdot|x)$.

Next, we evaluate the derivative of $w_{k'}^{\text{turn}}(\theta)$ with respect to the log probability. Recall that $w_{k'}^{\text{turn}}(\theta)$ is defined as

$$
\begin{aligned}
w_{k'}^{\text{turn}}(\theta) &= \left( \frac{\pi_\theta(y^{k'}|x, y^{<k'})}{\pi_{\theta_{\text{old}}}(y^{k'}|x, y^{<k'})} \right)^{\frac{1}{|y^{k'}|}} \\
&= \left( \prod_{t=t_{k'}^{\text{start}}}^{t_{k'}^{\text{end}}} \frac{\pi_\theta(y_t|x, y_{<t})}{\pi_{\theta_{\text{old}}}(y_t|x, y_{<t})} \right)^{\frac{1}{|y^{k'}|}} \\
&= \exp\left( \frac{1}{|y^{k'}|} \sum_{t=t_{k'}^{\text{start}}}^{t_{k'}^{\text{end}}} [\log \pi_\theta(y_t|x, y_{<t}) - \log \pi_{\theta_{\text{old}}}(y_t|x, y_{<t})] \right) \\
&= \exp\left( -\frac{1}{|y^{k'}|} \sum_{t=t_{k'}^{\text{start}}}^{t_{k'}^{\text{end}}} \log \pi_{\theta_{\text{old}}}(y_t|x, y_{<t}) \right) \cdot \exp\left( \frac{1}{|y^{k'}|} \sum_{t=t_{k'}^{\text{start}}}^{t_{k'}^{\text{end}}} \log \pi_\theta(y_t|x, y_{<t}) \right), \quad (14)
\end{aligned}
$$

where the first exponential term in Eq. 14 is constant with respect to $\theta$. We can write $w_{k'}^{\text{turn}}(\theta) = \kappa \cdot \exp\left( f(\theta) \right)$ where

$$
\kappa = \exp\left( -\frac{1}{|y^{k'}|} \sum_{t=t_{k'}^{\text{start}}}^{t_{k'}^{\text{end}}} \log \pi_{\theta_{\text{old}}}(y_t|x, y_{<t}) \right),
$$

$$
f(\theta) = \frac{1}{|y^{k'}|} \sum_{t=t_{k'}^{\text{start}}}^{t_{k'}^{\text{end}}} \log \pi_\theta(y_t|x, y_{<t}). \quad (15)
$$

Consequently, the derivative of $w_{k'}^{\text{turn}}(\theta)$ with respect to the log probability is evaluated as

$$
\frac{\partial w_{k'}^{\text{turn}}(\theta)}{\partial \log \pi_\theta(y_{t'}|x, y_{<t'})} = \kappa \cdot \exp\left( f(\theta) \right) \cdot \frac{\partial f(\theta)}{\partial \log \pi_\theta(y_{t'}|x, y_{<t'})} \quad (16)
$$

$$
= w_{k'}^{\text{turn}}(\theta) \cdot \frac{\partial}{\partial \log \pi_\theta(y_{t'}|x, y_{<t'})} \left[ \frac{1}{|y^{k'}|} \sum_{t=t_{k'}^{\text{start}}}^{t_{k'}^{\text{end}}} \log \pi_\theta(y_t|x, y_{<t}) \right]
$$

$$
= w_{k'}^{\text{turn}}(\theta) \cdot \frac{1}{|y^{k'}|} \cdot \frac{\partial \log \pi_\theta(y_{t'}|x, y_{<t'})}{\partial \log \pi_\theta(y_{t'}|x, y_{<t'})} \quad (17)
$$

$$
= w_{k'}^{\text{turn}}(\theta) \cdot \frac{1}{|y^{k'}|}, \quad (18)
$$

where Eq. 16 follows from the chain rule; and Eq. 17 follows since only the term with index $t = t'$ in the summation depends on $\log \pi_\theta(y_{t'}|x, y_{<t'})$. Substituting Eq. 18 back into Eq. 13, we get

$$
\frac{\partial \mathcal{J}_{\text{Turn-PPO}}(\theta)}{\partial \log \pi_\theta(y_{t'}|x, y_{<t'})} = \mathbb{E}\left[ \frac{w_{k'}^{\text{turn}}(\theta)}{|y| \cdot |y^{k'}|} \sum_{t=t_{k'}^{\text{start}}}^{t_{k'}^{\text{end}}} \mathbb{1}_{(k,t) \in \mathcal{B}_{\text{turn}}} \hat{A}_t \right]. \quad (19)
$$

Finally, from Eq. 11, the full gradient is given by

$$\nabla_\theta \mathcal{J}_{\text{Turn}-\text{PPO}}(\theta) = \sum_{k=1}^{K} \sum_{t=t_k^{\text{start}}}^{t_k^{\text{end}}} \frac{\partial \mathcal{J}_{\text{Turn}-\text{PPO}}(\theta)}{\partial \log \pi_\theta(y_t|x, y_{<t})} \nabla_\theta \log \pi_\theta(y_t|x, y_{<t}) \tag{20}$$

$$= \sum_{k=1}^{K} \sum_{t=t_k^{\text{start}}}^{t_k^{\text{end}}} \mathbb{E}\left[ \frac{w_k^{\text{turn}}(\theta)}{|y| \cdot |y^k|} \sum_{j=t_k^{\text{start}}}^{t_k^{\text{end}}} \mathbb{1}_{(k,j)\in\mathcal{B}_{\text{turn}}} \hat{A}_j \right] \nabla_\theta \log \pi_\theta(y_t|x, y_{<t}) \tag{21}$$

$$= \mathbb{E}\left[ \sum_{k=1}^{K} \frac{w_k^{\text{turn}}(\theta)}{|y| \cdot |y^k|} \left( \sum_{j=t_k^{\text{start}}}^{t_k^{\text{end}}} \mathbb{1}_{(k,j)\in\mathcal{B}_{\text{turn}}} \hat{A}_j \right) \sum_{t=t_k^{\text{start}}}^{t_k^{\text{end}}} \nabla_\theta \log \pi_\theta(y_t|x, y_{<t}) \right]$$

$$= \mathbb{E}\left[ \sum_{k=1}^{K} \frac{w_k^{\text{turn}}(\theta)}{|y| \cdot |y^k|} \left( \sum_{j=t_k^{\text{start}}}^{t_k^{\text{end}}} \mathbb{1}_{(k,j)\in\mathcal{B}_{\text{turn}}} \hat{A}_j \right) \nabla_\theta \log \pi_\theta(y^k|x, y^{<k}) \right] \tag{22}$$

$$= \mathbb{E}\left[ \frac{1}{|y|} \sum_{k=1}^{K} w_k^{\text{turn}}(\theta) \frac{\hat{A}^k}{|y^k|} \nabla_\theta \log \pi_\theta(y^k|x, y^{<k}) \right], \tag{23}$$

where Eq. 20 applies the multivariable chain rule; Eq. 21 follows from Eq. 19; Eq. 22 uses the chain rule identity $\nabla_\theta \log \pi_\theta(y^k|x, y^{<k}) = \sum_{t=t_k^{\text{start}}}^{t_k^{\text{end}}} \nabla_\theta \log \pi_\theta(y_t|x, y_{<t})$; and Eq. 23 follows by defining $\hat{A}^k \coloneqq \sum_{j=t_k^{\text{start}}}^{t_k^{\text{end}}} \mathbb{1}_{(k,j)\in\mathcal{B}_{\text{turn}}} \hat{A}_j$. This completes the proof of Lemma 4.1. ∎

### A.3 PROOF OF LEMMA 4.2

The token-level PPO objective function can be expressed as

$$\mathcal{J}_{\text{PPO}}(\theta) = \mathbb{E}_{x\sim D,\ y\sim\pi_{\theta_{\text{old}}}(\cdot|x)} \left[ \frac{1}{|y|} \sum_{t=1}^{|y|} \min(w_t(\theta)\hat{A}_t, \text{clip}(w_t(\theta), 1-\epsilon, 1+\epsilon)\hat{A}_t) \right]$$

$$= \mathbb{E}_{x\sim D,\ y\sim\pi_{\theta_{\text{old}}}(\cdot|x)} \left[ \frac{1}{|y|} \sum_{t=1}^{|y|} \left( \mathbb{1}_{t\in\mathcal{B}_{\text{token}}} \cdot w_t(\theta)\hat{A}_t + \mathbb{1}_{t\in\mathcal{B}_{\text{token}}^c} \cdot \text{clip}(w_t(\theta), 1-\epsilon, 1+\epsilon)\hat{A}_t \right) \right], \tag{24}$$

where Eq. 24 follows from the definition of the event $\mathcal{B}_{\text{token}}$ and

$$\text{clip}(w, 1-\epsilon, 1+\epsilon) = \max(\min(w, 1+\epsilon), 1-\epsilon) = \begin{cases} 1-\epsilon & \text{if } w < 1-\epsilon, \\ w & \text{if } 1-\epsilon \le w \le 1+\epsilon, \\ 1+\epsilon & \text{if } w > 1+\epsilon. \end{cases} \tag{25}$$

Taking the gradient with respect to $\theta$, we obtain

$$\nabla_\theta \mathcal{J}_{\text{PPO}}(\theta) = \mathbb{E}\left[ \frac{1}{|y|} \sum_{t=1}^{|y|} \left( \mathbb{1}_{t\in\mathcal{B}_{\text{token}}} \cdot \nabla_\theta w_t(\theta)\hat{A}_t + \mathbb{1}_{t\in\mathcal{B}_{\text{token}}^c} \cdot \nabla_\theta \text{clip}(w_t(\theta), 1-\epsilon, 1+\epsilon)\hat{A}_t \right) \right]$$

$$= \mathbb{E}\left[ \frac{1}{|y|} \sum_{t=1}^{|y|} \mathbb{1}_{t\in\mathcal{B}_{\text{token}}} \cdot \nabla_\theta w_t(\theta)\hat{A}_t \right] \tag{26}$$

$$= \mathbb{E}\left[ \frac{1}{|y|} \sum_{t=1}^{|y|} \mathbb{1}_{t\in\mathcal{B}_{\text{token}}} w_t(\theta)\nabla_\theta \log \pi_\theta(y_t|x, y_{<t})\hat{A}_t \right], \tag{27}$$

where Eq. 26 follows from the definition of the event $B^c$ that yields

$$\mathbb{1}_{t\in\mathcal{B}_{\text{token}}^c} \cdot \nabla_\theta \text{clip}(w, 1-\epsilon, 1+\epsilon)\hat{A}_t = \begin{cases} \nabla_\theta(1-\epsilon)\hat{A}_t = 0 & \text{if } \hat{A}_t < 0 \text{ and } w_t < 1-\epsilon, \\ \nabla_\theta(1+\epsilon)\hat{A}_t = 0 & \text{if } \hat{A}_t \ge 0 \text{ and } w_t > 1+\epsilon; \end{cases} \tag{28}$$

and Eq. 27 readily follows from the definition of $w_t(\theta)$ that leads to

$$\nabla_\theta w_t(\theta) = \nabla_\theta \frac{\pi_\theta(y_t|x, y_{<t})}{\pi_{old}(y_t|x, y_{<t})} = w_t(\theta)\nabla_\theta \log \pi_\theta(y_t|x, y_{<t}). \quad (29)$$

Finally, we verify that RHS of Lemma 4.2 is equal to Eq. 27 as follows:

$$\mathbb{E}\left[\frac{1}{|y|}\sum_{t=1}^{|y|} w_t\nabla_\theta \log \pi_\theta A_t^\pi\right] + \mathbb{E}\left[\frac{1}{|y|}\sum_{t=1}^{|y|} w_t\nabla_\theta \log \pi_\theta(\hat{A}_t - A_t^\pi)\right] - \mathbb{E}\left[\frac{1}{|y|}\sum_{t=1}^{|y|} \mathbb{1}_{t\in\mathcal{B}_{\text{token}}^c} w_t\nabla_\theta \log \pi_\theta \hat{A}_t\right]$$

$$= \mathbb{E}\left[\frac{1}{|y|}\sum_{t=1}^{|y|} w_t\nabla_\theta \log \pi_\theta \hat{A}_t\right] - \mathbb{E}\left[\frac{1}{|y|}\sum_{t=1}^{|y|} \mathbb{1}_{t\in\mathcal{B}_{\text{token}}^c} w_t\nabla_\theta \log \pi_\theta \hat{A}_t\right]$$

$$= \mathbb{E}\left[\frac{1}{|y|}\sum_{t=1}^{|y|} \mathbb{1}_{t\in\mathcal{B}_{\text{token}}} w_t\nabla_\theta \log \pi_\theta \hat{A}_t\right], \quad (30)$$

where Eq. 30 follows from $(1 - \mathbb{1}_{t\in\mathcal{B}_{\text{token}}^c}) = \mathbb{1}_{t\in\mathcal{B}_{\text{token}}}$. This completes the proof of Lemma 4.2. ∎

### A.4  IMPLEMENTATION DETAILS

**Retrieval System.** A local retriever service is deployed and accessed via an HTTP interface. For each user query, we consistently return the top three passages. The dialogue is restricted to a maximum of three interaction turns.

**Training Process.** All experiments are run on 8 NVIDIA H100 GPUs. We activate gradient checkpointing and train under a Fully Sharded Data Parallel (FSDP) setup with offloading enabled for parameters, gradients, and optimizers. The policy network is optimized with a learning rate of $1 \times 10^{-6}$, while the critic is trained with $1 \times 10^{-5}$. Optimization proceeds for 4 epochs, with warm-up ratios of 0.285 (policy) and 0.015 (critic). The effective batch size is 512, subdivided into mini-batches of 256 and micro-batches of 64 for policy updates and 8 for critic updates. Generalized Advantage Estimation (GAE) is employed with $\lambda = 1$ and $\gamma = 1$. Input sequences are truncated to at most 4,096 tokens, with limits of 500 tokens for responses, 2,048 tokens for the initial context, and 500 tokens for retrieved passages. We adopt turn-level importance sampling combined with variance-reduction detachment to improve training stability. Regularization follows PPO conventions with a KL coefficient of 0.001 and a clipping threshold of 0.2. Rollouts are generated using vLLM with tensor parallel size 4, GPU memory utilization set to 0.6, temperature fixed at 1.0, and top-$p$ sampling of 1.0. The rollout and reference log-probability calculations both use a micro-batch size of 128.

**Turn Boundary Identification.** To implement turn-level importance sampling, we need to identify turn boundaries within the multi-turn trajectories. In our search task setup, we use the loss mask to distinguish between agent-generated content and environment responses: LLM-generated reasoning and query formulation steps are marked with 1 in the `loss_mask`, while retrieved document content is marked with 0. We define turn boundaries by grouping consecutive tokens with `loss_mask = 1` as complete turns (corresponding to the agent's actions in our turn-level MDP), while consecutive 0s represent states (i.e., retrieved content that provides environmental feedback). These identified turn boundaries enable our algorithm to apply turn-level importance sampling and credit assignment.

### A.5  SUPPLEMENTARY EXPERIMENTS

In the following experiments, we set the training batch size to 512 and the mini-batch size to 128, which results in one on-policy update and three subsequent off-policy updates per batch. Under this more off-policy setting, our method demonstrates greater stability and more controlled performance compared to the baseline.

Figure 7 shows that the clipping bias grows steadily throughout training, indicating that the influence of off-policy data becomes increasingly significant. This growth is mainly driven by two factors: the variance of importance sampling ratios increases as the training distribution drifts away from the current policy, and the critic provides less accurate estimates on off-policy batches. As a result,

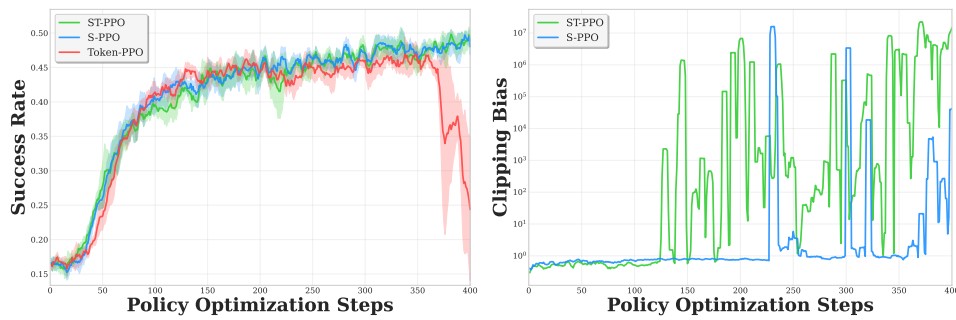

Figure 7: Experimental results of Qwen-2.5-7B policy models, with the value model also trained from Qwen-2.5-7B. Results are averaged over three trials. **Left:** Success Rate, **Right:** The norm of Clipping Bias.

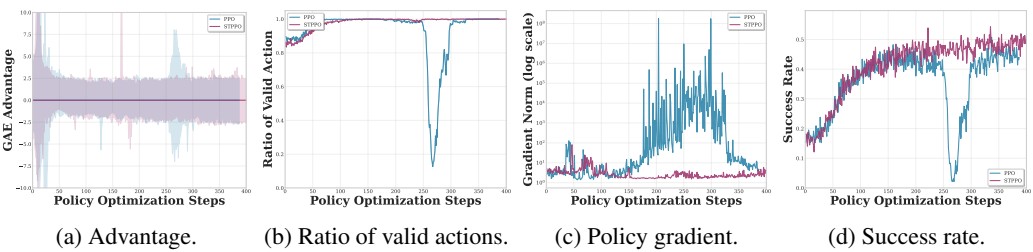

(a) Advantage.  (b) Ratio of valid actions.  (c) Policy gradient.  (d) Success rate.

Figure 8: Comparison between token-level PPO and ST-PPO on Qwen2.5-7B. Panels (a–d) report the estimated advantage, the ratio of valid actions, the L2 norm of the policy gradient, and the task success rate throughout training. Token-level PPO exhibits sharp spikes in gradient norm, collapses in valid-action ratio, and sudden drops in success rate, which indicate severe instability. In contrast, ST-PPO maintains stable advantages, smooth gradients, and consistently high task success rates across the entire training process.

the discrepancy between the true gradient and the clipped surrogate objective becomes amplified, leading to large and persistent bias in later stages of training. This observation highlights the necessity of employing the clipping ratio to reweight samples, as it counteracts the variance explosion of importance sampling and mitigates the critic's misestimation on off-policy data, thereby maintaining more stable optimization.

Additionally, in Figure 8, we compare the diagnostic signals of ST-PPO and token-level PPO. We observe that ST-PPO produces more stable advantage estimates and smoother gradient norms throughout training, whereas token-level PPO shows noticeable fluctuations. These steadier optimization signals in ST-PPO are accompanied by more consistent valid-action ratios and success rates, suggesting an overall improvement in training stability relative to token-level PPO.

### A.6 EXPERIMENTAL SETUP ON MEDICAL MULTIPLE-CHOICE BENCHMARKS

In this section, we further evaluate the performance of ST-PPO on medical tasks to assess its generalization ability in a domain distinct from open-domain search. Specifically, we use the AlphaMed19K (Liu et al., 2025) dataset for training, with Wikipedia serving as the retrieval source. To ensure consistency across experiments and datasets, we adopt the prompt template shown in Appendix A.7 for all training runs. The hyperparameter configuration for the medical dataset follows the same settings used in the NQ and HotpotQA experiments.

**Baselines.** We compare our method against several 8B-scale baselines commonly used in agentic tasks: (1) **RAG**: Retrieval-augmented generation (Glass et al., 2022; Li et al., 2025b) using the tool-following instructions, where external documents are retrieved to supplement the model's responses. Note that the Llama-3.1-8B-Instruct model is used without additional training in this setting. (2) **COT**: Chain-of-Thought prompting (Wei et al., 2022), which encourages the model to generate explicit reasoning steps before producing an answer. (3) **Search-R1**: Search-R1 framework Jin et al. (2025b),

Table 1: Performance of 8B medical LLMs across in-domain[†] and out-of-domain[*] benchmarks (accuracy %). Bold numbers indicate the best performance within each model family.

| Model | In-Domain[†] | | Out-of-Domain[*] | | | Avg. |
|---|---|---|---|---|---|---|
| | MedQA | MedMCQA | PubMedQA | MMLU-M | MedXpert | |
| *Inference-Based Methods (No Retrieval)* | | | | | | |
| Llama-3.1-8B-Instruct (Direct Inference) | 45.20 | 52.40 | 62.00 | 40.70 | 13.00 | 42.66 |
| Llama-3.1-8B-Instruct (CoT) | 48.62 | 59.80 | 64.40 | 54.20 | 13.02 | 48.01 |
| MedLlama-3-8B (CoT) | 66.60 | 53.40 | 64.40 | 45.70 | 11.04 | 48.23 |
| *Retrieval-Augmented and RL-Enhanced Methods* | | | | | | |
| Llama-3.1-8B-Instruct (RAG) | 8.90 | 11.30 | 16.80 | 4.90 | 9.20 | 10.22 |
| Llama-3.1-8B-Instruct (Search-R1) | 53.40 | 54.00 | 60.20 | 49.50 | 9.77 | 45.37 |
| Llama-3.1-8B-Instruct (ST-PPO, Ours) | **58.90** | **57.90** | **64.80** | **53.50** | **14.40** | **49.90** |

[†]In-domain benchmarks. [*]Out-of-domain benchmarks.

where we train the model using PPO and evaluate using the checkpoint at the 150th optimization step (2 epochs). (4) **ST-PPO**: Our proposed algorithm, evaluated using the checkpoint at the 150th step (2 epochs).

**Evaluation Datasets.** To comprehensively evaluate the algorithm's performance on medical tasks, we consider both in-domain and out-of-domain benchmarks. For in-domain evaluation, we use MedQA (Jin et al., 2020) and MedMCQA (Pal et al., 2022). For out-of-domain evaluation, we include PubMedQA (Jin et al., 2019), the medical subset of MMLU (Wang et al., 2024), and MedXpert (Zuo et al., 2025). We exclude GPQA (Rein et al., 2024) as it does not contain a medically relevant subset.

**Performance.** Table 1 shows that our training method not only stabilizes multi-turn optimization but also enables a model that initially lacks tool-use ability to reliably perform retrieval and acquire domain knowledge. The poor performance of RAG demonstrates this initial limitation. As a result, ST-PPO achieves the strongest overall performance on medical QA benchmarks, with the best average accuracy (49.90%) among retrieval-augmented and RL-enhanced methods, outperforming vanilla PPO in Search-R1 framework across both in-domain and out-of-domain evaluations.

### A.7 EXCLUSION RATIONALE FOR OFF-POLICY ALGORITHMS BASELINES

For existing off-policy RL methods, we do not include RePO (Li et al., 2025a) because their models are trained under a maximum response length of 1024 tokens. We do not include LUFFY (Yan et al., 2025) since the usage of off-policy guidance from a superior model (e.g., DeepSeek-R1), which is also viewed as a different setting where extrinsic guidance or demonstrations are accessible. We exclude Tapered Off-policy REINFORCE (Roux et al., 2025) since they didn't release their code or checkpoints.

> **Prefix Prompt for the Medical Task**
>
> Answer the given medical multiple choice question. Think step-by-step inside <think> and </think> tags.
> When you encounter:
>
> - Unfamiliar medical terminology or drug names
> - Complex disease mechanisms you're uncertain about
> - Specific treatment protocols or guidelines you need to verify
> - Any information where you lack confidence
>
> You can search for clarification using <search> query </search>, and results will appear between <information> and </information>. Use search strategically to fill knowledge gaps and improve answer accuracy. After sufficient reasoning and any necessary searches, provide your final answer inside <answer> and </answer> with ONLY the letter (e.g., <answer>A</answer>).

Table 2: A rollout example of answering a medical question.

**Question:** Which one of the following is not included in the treatment of malignant melanoma?

**Options:**
A: Radiation
B: Surgical excision
C: Chemotherapy
D: Immunotherapy

<think> Let's break down the question step by step! We're looking for the option that is not included in the treatment of malignant melanoma. </think>

<think> A: Radiation - Hmm, radiation therapy is sometimes used in combination with surgery or as palliative care for symptomatic relief. Not sure if it's the odd one out.
B: Surgical excision - Ah, surgery is a standard treatment for removing the primary tumor and any involved lymph nodes. Yes, that's a correct treatment option.
C: Chemotherapy - Chemotherapy can be used in some cases of malignant melanoma, particularly for patients with advanced disease or those who have recurrence after initial treatment. Yes, that's a valid treatment option.
D: Immunotherapy - Immunotherapy, such as pembrolizumab (Keytruda) or nivolumab (Opdivo), has shown significant efficacy in treating advanced melanoma by enhancing the body's immune response against cancer cells. Yes, that's a commonly used treatment option!
I'm getting confused about the radiation option, so let me search for clarification. </think>

<search> What is the role of radiation therapy in treating malignant melanoma? </search>

<information> Doc 1 (Melanoma): Radiation therapy is often used after surgical resection for locally or regionally advanced or unresectable metastases. It may reduce local recurrence but does not prolong survival. Doc 2 (Radiation therapy): Melanoma is generally radioresistant; radiation is mainly palliative for metastatic disease. Doc 3 (Radiation therapy): Radiation requires higher doses for melanoma than most cancers and is not curative. </information>

<think> Ah-ha! It seems that radiation therapy has a limited role in treating malignant melanoma and is mostly used in palliative care settings. It's not the first-line treatment for the disease. So, I think I can confidently say that... </think>

<answer> A </answer>

