# OpenReview forum: "ST-PPO: Stabilized Off-Policy Proximal Policy Optimization for Multi-Turn Agents"
_ICLR.cc/2026/Conference — Submitted to ICLR 2026_

### Official Review · Reviewer_apVh · 2025-10-28

**Soundness:** 2
**Presentation:** 4
**Contribution:** 2
**Rating:** 2
**Confidence:** 4

**Summary:**

The paper introduces new techniques to stabilise training of LLMs using PPO which are motivated by trying to solve empirically identified problems. These techniques are turn-level importance sampling and clipping-bias correction. These should respectively better represent the turn-based nature of LLM chat finetuning, and reduce the variance in gradient updates from off-policy samples. Empirical analysis shows these techniques provide better performance on multi-turn tasks compared to standard PPO.

**Strengths:**

1. The problem of variance in the GAE estimation causing issues is well evidenced by prior work and empirical experiments within the paper, and so the introduction of a method specifically to mitigate this is well-motivated
2. The exposition of the paper is clear, and the method is well explained
3. Results clearly show that the clipping-bias correction technique improves performance relative to vanilla PPO
4. The decompositions of loss functions and the relating lemmas are nice additions that improve the clarity of the arguments being made in the paper

**Weaknesses:**

1. The method is only compared with PPO, and not other competitive LLM RL algorithms such as GRPO. Additionally, it is not compared to cited works seeking to solve similar problems such as TOPR, LUFFY, and GSPO.
2. There does not seem to be sufficient evidence to conclude that the turn-level structure specifically is causing issues with PPO, weakening the motivation for some of the techniques proposed in the paper.
3. In figure 3a there is heavy overlap between the error bars for turn and token PPO, there is not enough evidence here to conclude turn-PPO outperforms token-PPO.
4. Figure 3d does not fully support the claim that the L2 norm of the clipping bias term is growing exponentially, as the data is extremely noisy and no linear trend has been fitted, let alone had its goodness of fit computed.
5. ST-PPO and S-PPO perform similarly, providing additional evidence that the influence of accounting for turns is minimal. In fact, figure 4b shows S-PPO outperforming ST-PPO.

**Questions:**

Questions:
1. Why did you not test against other baselines beyond vanilla PPO?
2. Do you have any other evidence that one either ought to expect turn-PPO to work better than token-PPO, or that turn-PPO does in fact work better than token-PPO?
3. Could you provide a (smoothed) version of figure 3d with a trend line plotted?

Suggestions:
Abandon the turn-based component and focus on the clipping-bias correction. Test this against more baselines and on a wider range of benchmarks. Test if analogous methods improve algorithms such as GRPO or RLOO, as alluded to being possible on lines 398-400.

---

> ### Author Response · Authors · 2025-12-02
>
> We thank the reviewer for their detailed feedback and for highlighting that our method is well-motivated by prior work and our empirical evidence of variance issues in GAE estimation. We particularly appreciate the reviewer's acknowledgment of the clear exposition and explanation of our method, the demonstrated performance improvements of our clipping-bias correction technique over vanilla PPO, and the contribution of our loss function decompositions and lemmas to the clarity of our arguments. Our responses to the specific weaknesses, questions, and suggestions are provided below.
>
> ### Responses to Weaknesses
>
> **Note**: Responses to W.2 & W.4 are provided in the following comment.
>
> > W.1. **Only compared with PPO; missing GRPO, TOPR, LUFFY, GSPO.**
>
> **Response:** Thank you for the comment. Some of the mentioned baselines cannot be fairly or feasibly included:
>
> 1. TOPR does not provide released code or checkpoints, making reliable reproduction difficult.
> 3. LUFFY requires off-policy guidance from a stronger teacher model (e.g., DeepSeek-R1), which corresponds to a different learning setting.
> 3. RePO is trained under a strict 1024-token response limit, which is incompatible with our long-context multi-turn setup.
>
> To address your concern, we have additionally included comparisons against GRPO and GSPO in the revised version in **Figure 6**. These results show that our proposed method continues to outperform these stronger baselines, further confirming the effectiveness of our approach.
>
> > W.3. **Error bars in Figure 3 overlap heavily; cannot conclude turn-PPO > token-PPO.**
>
> **Response:** Thank you for pointing this out. We agree that the performance gap in Figure 3(a) is small and the error bars overlap, so no strong conclusion should be drawn there.
>
> Our intention was not to claim turn-PPO > token-PPO, but to highlight a trend: Figure 3(b) shows that turn-PPO yields **smaller gradient norms**, which motivated the stabilized variant ST-PPO. As shown in Figure 5, **ST-PPO achieves the lowest gradient norms overall**, indicating that the stabilization effect becomes clearer at larger model scales.
>
> Turn-PPO is therefore presented as an intermediate variant illustrating useful structure, while the main stabilization benefits come from S-PPO and ST-PPO.
>
> > W.5. **ST-PPO ≈ S-PPO; Figure 4b even shows S-PPO outperforming ST-PPO.**
>
> **Response:** Thank you for the observation. In our paper, we do not aim to claim that ST-PPO is strictly better than S-PPO, or vice versa. These two methods are both part of our proposed framework, each addressing instability from a different angle.
>
> As seen in Figure 4b, S-PPO may appear slightly better in that particular setting, while in other cases ST-PPO performs comparably or marginally stronger. Given the natural variance across tasks and runs, we do not view these differences as statistically conclusive.
>
> The intended takeaway is that both variants substantially stabilize training compared to standard PPO, and ST-PPO simply provides a unified option that combines turn-level sampling with clipping-bias correction for users who prefer an integrated design.

---

> ### Author Response · Authors · 2025-12-02
>
> ### Responses to Questions
> > Q.1. **Why did you not test against other baselines beyond vanilla PPO?**
>
> **Response:** To address your concern, we have additionally included comparisons against GRPO and GSPO in the revised version in both **Figure 6** and Appendix A.6. These results show that our proposed method continues to outperform these stronger baselines.
>
> > - W.2. **Insufficient evidence that turn-level structure specifically causes PPO instability.**
> > - Q.2. **Is there any evidence that turn-PPO should outperform token-PPO?**
>
> **Response:** Turn-level PPO serves mainly as an intermediate variant in our framework. While it already mitigates the granularity mismatch in multi-turn reasoning, its primary value is that it motivated and revealed the gradient-norm explosion phenomenon under off-policy updates. This insight directly inspired our further development of the clipping-bias correction, ultimately leading to the full ST-PPO algorithm.
>
> > - W.4. **Figure 3d does not fully support the claim that the L2 norm of the clipping bias term is growing exponentially, as the data is extremely noisy and no linear trend has been fitted, let alone had its goodness of fit computed.**
> > - Q.3. **Could you provide a smoothed version of Figure 3d with a trend line?**
>
> **Response:** Thank you for pointing this out. We would like to clarify that Figure 3d is already plotted with smoothing, precisely because the raw clipping-bias values are extremely noisy in practice. This level of noisiness is expected: the clipping-bias term has no inherent regularity, and once off-policy samples accumulate, it often exhibits sudden large spikes that do not return to a stable range afterward.
>
> Our intention was not to claim a strict exponential law, but rather to highlight the following qualitative behavior: the clipping-bias norm grows sharply after a certain point, and once such a spike occurs, it typically leads to unstable policy updates, since samples with very large clipping bias are unreliable for optimization.
>
> To avoid misunderstanding, we have updated the description for Figure 3(d).
>
> ---
>
> ### Responses to Suggestions
> > S.1. **Abandon the turn-based component and focus on the clipping-bias correction. Test this against more baselines and on a wider range of benchmarks. Test if analogous methods improve algorithms such as GRPO or RLOO, as alluded to being possible on lines 398-400.**
>
> **Response:** Thank you for the thoughtful suggestion. We agree that clipping-bias correction plays a central role in stabilizing training. To verify this, in the revised version, we introduced a new variant S-GRPO, which applies the same correction to the GRPO framework. As shown in the updated experiments, S-GRPO significantly improves stability over both GRPO and GSPO, demonstrating that the correction mechanism is broadly applicable beyond PPO-style objectives.
>
> At the same time, we chose to keep the turn-based component in the paper because it represents an important step in our experimental process. Even though the empirical gains from Turn-PPO are not strong, this exploration helped us better understand how gradient-norm issues and credit-assignment mismatches contribute to instability. We believe documenting this progression can offer useful insights to readers and potentially spark further ideas about how turn structure and credit assignment interact in multi-turn settings.

---

### Official Review · Reviewer_pNPG · 2025-11-01

**Soundness:** 2
**Presentation:** 3
**Contribution:** 2
**Rating:** 4
**Confidence:** 4

**Summary:**

This paper addresses the instability and performance collapse of Proximal Policy Optimization (PPO) when used for training multi-turn LLM agents. The authors identify two primary sources of this instability: (1) a mismatch between token-level importance sampling and the natural turn-level structure of agentic tasks, and (2) high-variance gradients resulting from inaccurate critic estimates on off-policy samples. To solve this, they propose two techniques: **turn-level importance sampling** to align credit assignment with task granularity, and a **clipping-bias correction** to normalize gradients and downweight unreliable samples. The combined algorithm, ST-PPO, is shown empirically to prevent performance collapse on multi-turn search tasks and achieve higher final task performance compared to standard token-level PPO.

**Strengths:**

- **Important Problem:** The paper addresses a critical and practical challenge. PPO is a foundational algorithm for LLM alignment, and its instability in multi-turn, off-policy settings is a significant bottleneck for developing more complex reasoning agents.
- **Clear Diagnosis:** The paper does a good job of empirically diagnosing the root causes of instability, specifically linking gradient spikes and performance collapse to unreliable advantage estimates from the critic on off-policy data.
- **Sufficient Literature Review:** The work is well-contextualized within existing research on RLHF, off-policy learning for LLMs, and multi-turn agent training.
- **Clarity:** The paper is generally well-written and clearly motivates its proposed solutions from its analysis of the problem.

**Weaknesses:**

- **Empirical Gaps:** The empirical evaluations, while showing the final methods are stable, could be more robust in demonstrating the distinct value of each proposed component.

  - In Figure 3a, the performance improvement of **Turn-PPO** (turn-level sampling only) over token-level PPO is minor.
  - In Figure 4, the performance curves for **S-PPO** (clipping-bias correction only) and **ST-PPO** (both components) are nearly identical. This makes it difficult to assess the added benefit of the turn-level importance sampling when the clipping-bias correction is already in use.
  - The diagnostic plots in Figure 3(c-d) (PPO loss norm and clipping bias norm) are only provided for the failing token-level PPO. These plots are crucial for the paper's argument, and not showing them for S-PPO and ST-PPO is a missed opportunity to validate that the proposed correction mechanism is working as intended.

- **Limited Baselines and Models:** The experimental comparison is somewhat limited.

  - The main stability plots in Figure 4 compare S-PPO and ST-PPO only against the standard Token-PPO, which collapses. Other relevant baselines are not included in this comparison.

  - The experiments are conducted on a single model family (Qwen-2.5-7B and 1.5B). Demonstrating the method's effectiveness on more diverse model architectures would significantly strengthen the paper's claims.

**Questions:**

1. Could the authors provide more intuition behind the clipping-bias correction in Equations 7 and 8? Why does normalizing the gradient by the L2 norm of the *clipping bias term* ($C(\theta)$)  effectively downweight unreliable samples and lead to stabilization?

2. The paper presents the two techniques as complementary. However, the results in Figure 4 suggest that S-PPO and ST-PPO are empirically indistinguishable. Does this imply that the clipping-bias correction is the dominant and perhaps sufficient stabilization technique, and that turn-level sampling offers little additional benefit *after* this correction is applied? What is the specific interplay between these two modules?

---

> ### Author Response · Authors · 2025-12-02
>
> We gratefully thank the reviewer for their thoughtful feedback and for acknowledging our clear empirical diagnosis and our thorough literature review. Below, we address each weakness and question raised.
>
> ### Responses to Weaknesses
>
> **Empirical Gaps:**
>
> > W.1.1. **Improvement of Turn-PPO over token-PPO is minor (Figure 3a).**
>
> **Response:** Thank you for this observation. We agree that the improvement of Turn-PPO over token-level PPO is modest. As noted in the paper, Turn-PPO gives a small benefit on smaller models like Qwen2.5-1.5B, but it does not work well on larger models such as Qwen2.5-7B. It was not meant to be a central contribution but an early exploratory variant.
>
> Even so, it helped clarify the broader picture. Turn-level sampling reduces the granularity mismatch, but this alone cannot address the main source of instability. Much of the variance comes from off-policy tokens receiving unreliable credit, which leads to noisy advantages and occasional gradient spikes.
>
> In this sense, Turn-PPO served as a useful stepping stone. It pointed us toward the need to address both granularity and off-policy variance at the same time. This is why the final focus of the paper is on the stabilized variants, especially ST-PPO, which remain stable even at larger model scales.
>
> > W.1.2. **Performance curves for S-PPO and ST-PPO in Figure 4 are nearly identical, making added benefit of turn-level sampling unclear.**
>
>  **Response:** Thank you for the observation. Our goal is not to claim that turn-level sampling must produce a large performance gap between S-PPO and ST-PPO. In fact, both methods already address the main source of instability—the clipping-bias explosion—and therefore both achieve substantially more stable training than token-level PPO.
>
> The added effect of turn-level sampling in ST-PPO is more conservative gradient behavior, which we highlight through consistently lower gradient norms (Figures 4 and 5). While this additional conservativeness does not always translate into a visible performance gap in final success rates, it leads to smoother optimization and reduces the chance of rare but catastrophic gradient spikes.
>
> For these reasons, we view S-PPO and ST-PPO as complementary variants that both successfully stabilize multi-turn PPO training. Importantly, all three variants (Turn-PPO, S-PPO, and ST-PPO) are methods proposed in this work, and all of them achieve clearly better stability and performance compared to the baseline token-level PPO. To clarify their distinctions, we have added a table summarizing the three proposed variants and the specific issue each one addresses.
>
> | Method      | Importance Sampling (IS) Granularity | Credit Assignment Granularity | Gradient Stability Mechanism |
> |-------------|--------------------------------|-------------------------------|-------------------------------|
> | **Token-PPO** | Token-level IS ratio $w_t$ | Token-level credit | No bias correction |
> | **Turn-PPO**  | Turn-level IS ratio $w^{\text{turn}}_k$ | Aggregated turn-level credit  | No bias correction |
> | **S-PPO**     | Token-level IS ratio $w_t$ | Token-level credit | Gradient normalization via $\|C_{\text{token}}(\theta)\|_2^{-1}$ |
> | **ST-PPO**    | Turn-level IS ratio $w^{\text{turn}}_k$ | Aggregated turn-level credit  | Gradient normalization via $\|C_{\text{turn}}(\theta)\|_2^{-1}$ |
>
> > W.1.3. **Figure 3(c–d) only shows diagnostics for failing token-PPO; missing S-PPO/ST-PPO diagnostics.**
>
> **Response:** Thank you for pointing this out. To address the reviewer’s concern, we have added the corresponding diagnostic curves for both S-PPO and ST-PPO in **Figure 8** in the appendix, showing that their clipping-bias norms and gradient behaviors remain consistently stable throughout training.
>
> **Limited Baselines and Models:**
> > - W.2.1. **The main stability plots in Figure 4 compare S-PPO and ST-PPO only against the standard Token-PPO, which collapses. Other relevant baselines are not included in this comparison.**
> > - W.2.2 **The experiments are conducted on a single model family (Qwen-2.5-7B and 1.5B). Demonstrating the method's effectiveness on more diverse model architectures would significantly strengthen the paper's claims.**
>
> **Response:** Thank you for the comment. We have addressed both concerns (baseline coverage and model diversity) in the revised version.
>
> **Expanded baselines:** We have added comparisons against GRPO and GSPO on the NQ dataset, together with our S-GRPO variant. As shown in the newly added results (Figure 6), both ST-PPO and S-PPO achieve more stable training than vanilla PPO, while S-GRPO further improves stability over GRPO and GSPO.
>
> **Model diversity:** We also expanded our experiments beyond Qwen-2.5 by adding results on a medical QA task using the Llama3-8B-Instruct model. The new experiments (Appendix A.6–A.8) show that our stabilization approach continues to yield strong and consistent improvements in a different domain and under a different model architecture.

---

> ### Author Response · Authors · 2025-12-02
>
> ### Responses to Questions
>
> > Q.1. **Could the authors provide more intuition behind the clipping-bias correction (Equations 7 and 8)? Why does L2 normalization stabilize gradients?**
>
> **Response:**
> Thank you for the insightful question. The key intuition behind the clipping-bias correction is the mismatch between the clipping mechanism and the granularity of credit assignment.
>
> In the original PPO paper [1], clipping is applied at the **state–action** level: if an entire action falls outside the trust region, the gradient for that state–action pair is fully suppressed. In this setting, the clipping mechanism and the credit assignment are perfectly aligned.
>
> However, in multi-turn LLMs, the **decision is made at the turn level**, while both **advantages and importance ratios are computed at the token level**. This mismatch can lead to risky updates. For example, if only a few key tokens determine the intent of a generation but other masked or off-policy tokens dominate the update, the model may receive an unstable or semantically inconsistent learning signal. Our motivation is therefore to downweight such unreliable samples through the clipping-bias correction. This also encourages a more conservative update: when a sample is highly off-policy, its gradient contribution is reduced, making the algorithm behave closer to an on-policy method.
>
> The clipping-bias term
> $$
> C(\theta)=\mathbb{E}\Big[\frac{1}{|y|}\sum_{t\in B^{c}} w_t \hat A_t\Big]
> $$  captures the **off-policyness** of a sample—i.e., the extent to which its tokens lie outside the trust region and contribute unreliable updates. Normalizing the gradient by $\|C(\theta)\|_2$ (Eqs. 7–8) creates a **soft filtering effect**: samples with higher off-policyness are automatically downweighted, mimicking PPO’s original action-wise clipping behavior. This restores a turn-level trust-region effect and suppresses unstable updates, leading to much more stable optimization in multi-turn settings.
>
> Reference: [1] Schulman, John, et al. "Proximal policy optimization algorithms." arXiv preprint arXiv:1707.06347 (2017).
>
> > Q.2. **S-PPO and ST-PPO are empirically indistinguishable—does turn-level sampling offer additional benefit after clipping-bias correction?**
>
> **Response:** Thank you for the question. Based on our empirical observations, both S-PPO and ST-PPO are highly effective at stabilizing training. This strong stabilization effect is likely why the two methods appear empirically similar: once clipping-bias correction removes most of the variance-induced instability, the additional benefit from turn-level sampling becomes less visually distinguishable.
>
> Indeed, as shown in Figure 4, both algorithms exhibit nearly identical convergence behavior and achieve consistently strong performance. In practice, this confirms that clipping-bias correction is the dominant factor for stability, while turn-level sampling provides complementary but smaller gains under already-stable conditions.

---

### Official Review · Reviewer_3C9c · 2025-11-01

**Soundness:** 3
**Presentation:** 3
**Contribution:** 2
**Rating:** 4
**Confidence:** 3

**Summary:**

This paper addresses the instability and performance collapse of Proximal Policy Optimization (PPO) when training large language models (LLMs) on multi-turn tasks. The authors identify two root causes for this instability: a "granularity mismatch" between PPO's token-level optimization and turn-level credit assignment, and high-variance gradients from off-policy critic estimates. To solve this, they introduce ST-PPO, an algorithm featuring two new techniques: turn-level importance sampling, which aligns credit assignment with the multi-turn structure, and clipping-bias correction, which normalizes gradients by downweighting unreliable, highly off-policy samples. Experiments on multi-turn search tasks demonstrate that ST-PPO and its variants successfully prevent performance collapses, maintain greater stability, and achieve higher task performance than standard token-level PPO.

**Strengths:**

- This paper proposes turn-level importance sampling, a novel technique that aligns the RL optimization process with the natural turn-based structure of dialogue and reasoning tasks.
- This paper proposes clipping-bias correction, a stabilization mechanism that adaptively downweights unreliable, highly off-policy samples to prevent them from destabilizing gradient updates.

**Weaknesses:**

- This paper does not test on more up-to-date models, such as Qwen 3 series.
- The experiments primarily contrast ST-PPO with a standard, unstable token-level PPO. The paper does not include empirical comparisons against other advanced stabilization algorithms mentioned in its own related work (like TOPR, GSPO, or ARPO). This paper should include other PPO stabilization algorithms into baselines.
- The core "turn-level" component relies on a specific implementation to identify turn boundaries.

**Questions:**

- In Figure 4 & 5, it does not include the turn-ppo, what is its performance here?
- Does this method work in GRPO? In GRPO, GRPO also has clipping and should suffer similar problems.

---

> ### Author Response · Authors · 2025-12-02
>
> We sincerely thank the reviewer for their review of our paper and for recognizing the novelty of our turn-level importance sampling technique and our clipping-bias correction mechanism. We particularly appreciate the reviewer's acknowledgment that our approach aligns the RL optimization process with the natural turn-based structure of dialogue and reasoning tasks, and that our stabilization mechanism effectively prevents off-policy samples from destabilizing gradient updates. In what follows, we provide point-by-point responses to each weakness and question raised.
>
> ### Responses to Weaknesses
>
> > W.1. **This paper does not test on more up-to-date models, such as Qwen 3 series.**
>
> **Response:** Thank you for the valuable suggestion. We acknowledge that testing on the latest Qwen 3 series would strengthen our empirical validation. Due to current computational constraints, conducting the full experimental protocol (multiple seeds, complete training curves, and comprehensive ablations) on Qwen 3 would require approximately one month of H100 cluster time, which unfortunately exceeds our available resources for this revision cycle.
>
> To address your concern and demonstrate the generalizability of our approach beyond the Qwen 2.5 family, we have conducted additional experiments on **Llama3-8B-Instruct**. As shown in the newly added **Appendix A.6**, the proposed algorithm outperforms the baselines on this alternative model family. These results provide evidence that our performance claims about the proposed algorithm are not architecture-specific and generalize across different LLM backbones.
>
> We agree that evaluating Qwen 3 would be valuable future work and will prioritize it in extended versions of this research.
>
> > W.2. **The experiments primarily contrast ST-PPO with a standard, unstable token-level PPO. The paper does not include empirical comparisons against other advanced stabilization algorithms mentioned in its related work (like TOPR, GSPO, or ARPO).**
>
> **Response:** Thank you for the valuable suggestion. As noted in **Appendix A.7**, several advanced stabilization baselines cannot be fairly included in our comparisons: **TOPR** has no released code or checkpoints; **RePO** is trained under a strict 1024-token limit that is incompatible with our long-context setting; and **LUFFY** relies on off-policy guidance from a stronger teacher model (i.e. DeepSeek-R1), which constitutes a different problem setup.
>
> To address your concern, we have included comparisons against **GRPO** and **GSPO** in Figure 6 in the revised version.
>
> > W.3. **The core "turn-level" component relies on a specific implementation to identify turn boundaries.**
>
> **Response:** Thank you for raising this concern. We want to clarify that the “turn-level” formulation itself is general and does not depend on any specific implementation for identifying turn boundaries.
>
> In the experiments section, we introduce one simple and reproducible approach for search tasks: using the loss mask to distinguish between agent-generated content (marked with 1) and environment responses (marked with 0). This is presented as a practical implementation choice for our specific setting, not as a requirement of the method. In fact, different task settings may identify these boundaries through different means, such as special tokens, tool call completions, or other task-specific markers that segment the interaction into meaningful turns.
>
> In any turn-level interaction setting, an agent naturally makes decisions based on the current state/observation $S_t$. Therefore, turn boundaries can be inferred directly from state transitions. Operationally, these can be identified by examining the difference between consecutive dialogue states $S_{t+1}$ and $S_t$, where each state encodes the full history up to that turn.

---

> ### Author Response · Authors · 2025-12-04
>
> ### Responses to Questions
>
> > Q.1. **In Figures 4 & 5, turn-PPO is missing. What is its performance here?**
>
> **Response:** Thank you for the question. As shown in Figure 3, we include the results of turn-level PPO, where we observe that turn-PPO indeed reduces the gradient norm and stabilizes training compared to token-PPO. This observation motivated our development of the stabilized variants (S-PPO and ST-PPO). However, when testing it with 7B models, we found that turn-PPO also tended to collapse on multi-turn tasks. After observing this behavior, we did not perform extensive repeated runs of this baseline.
>
> > Q.2. **Does this method work in GRPO? GRPO also has clipping and should suffer similar problems.**
>
> **Response:** Thank you for the question. Based on our verification, GRPO does indeed suffer from the same clipping-induced instability. In the newly added experiments in **Figure 6** of the revised version, we further compare GRPO, GSPO, and our proposed S-GRPO on the NQ dataset. The results clearly show that S-GRPO is substantially more stable than both vanilla GRPO and GSPO, exhibiting smooth learning curves and avoiding the abrupt collapse observed in the original methods.

---

### Author Response · Authors · 2025-12-02
**Global Response**

Dear AC,

Thank you for overseeing the review process of our submission. Since this year's discussion phase ended unexpectedly early and we were unable to receive reviewer follow-ups, we have prepared this concise summary of the main concerns and how our revised submission addresses them. We hope this facilitates your assessment and clarifies our contributions.

We sincerely thank all reviewers for their thorough and constructive feedback. We are encouraged that reviewers recognized the importance of addressing PPO instability in multi-turn settings, found our diagnosis of gradient spikes from off-policy critics to be well-evidenced, and acknowledged the novelty of our technical contributions (turn-level importance sampling and clipping-bias correction). In response, we have substantially revised the paper with new experiments across models and tasks, extended baseline comparisons, comprehensive diagnostic analyses, and improved exposition.

Below, we summarize the key reviewer concerns and our corresponding updates. We believe these revisions address all major points and significantly strengthen both the empirical validation and conceptual clarity of our work.

---

## **A. Empirical Validation: New Models, Tasks, and Baselines**

### **(1) Broader Evaluation across Models and Tasks**

To address concerns regarding model and task diversity, we added comprehensive experiments on Llama-3.1-8B-Instruct across medical QA benchmarks (MedQA, MedMCQA, PubMedQA, MMLU-Medical, MedXpert). Results demonstrate that ST-PPO achieves 49.90% average accuracy, outperforming vanilla PPO (Search-R1) by 4.5 points and succeeding where standard RAG fails (10.22% accuracy). Notably, ST-PPO also surpasses inference-based methods without retrieval, including the same base model with CoT prompting (48.01%) and the specialized MedLlama-3-8B model (48.23%). This validates that our stabilization method generalizes beyond the Qwen2.5 family to enable effective multi-turn RL optimization on models that initially lack tool-use capabilities, across both in-domain and out-of-domain evaluations. This is discussed in Appendix A.6 of our revision.

### **(2) Extension to Alternative RL Method (S-GRPO)**

Following reviewers' interest in GRPO-style baselines, we introduced S-GRPO, which extends our clipping-bias normalization fromPPO to GRPO. Experiments (Figure 6) show that S-GRPO achieves substantially improved stability over GRPO and GSPO, with markedly smoother loss-norm curves. This validates that our stabilization mechanism generalizes beyond the specific PPO implementation, broadening its applicability to alternative RL training paradigms. This is discussed in Section 4.2 and Section 5 of our revision.

### **(3) Comparative Diagnostic Analysis of Training Stability: ST-PPO vs PPO**

We added comprehensive diagnostic comparisons between ST-PPO and token-level PPO across four key metrics: advantage estimates, valid-action ratio, policy gradient norms, and success rate (Figure 8). Results show that token-level PPO exhibits sharp gradient spikes, valid-action collapse, and sudden success rate drops, while ST-PPO maintains stable advantages, smooth gradients, and consistently high task performance throughout training. This directly confirms that our method effectively prevents the severe instability observed in standard token-level PPO. This is discussed in Appendix A.5 of our revision.

### **(4) Additional Baseline Comparisons and Exclusion Justifications**

In response to requests for more comprehensive comparisons, we added experiments with GRPO and GSPO (Figure 6), and provided detailed justification for why other recent methods cannot be fairly included under our long-context, multi-turn setting: TOPR (no released code), RePO (incompatible 1024-token limit), and LUFFY (requires off-policy teacher guidance, constituting a different problem setup). We also developed S-GRPO as a stabilized GRPO variant (detailed in Point #2 above). This is discussed in Section 4.2, Section 5, and Appendix A.7 of our revision.

---

---

> ### Author Response · Authors · 2025-12-04
>
> ---
>
> ## **B. Clarification of Algorithmic Contributions**
>
> ### **From PPO → Turn-PPO → S-PPO → ST-PPO**
>
> Reviewers asked about the role of Turn-PPO and why we ultimately favor ST-PPO. We have revised our manuscript to clarify this algorithmic progression and the distinct contributions of each variant:
>
> - **PPO** suffers from two issues in multi-turn LLM training: (1) granularity mismatch between token-level optimization and turn-level task structure, and (2) high variance from off-policy samples with critic estimates.
> - **Turn-PPO** addresses the first issue by aligning importance sampling with turn-level boundaries, reducing gradient norms (Figure 3b) but still exhibiting collapse when scaled to larger models such as Qwen2.5-7B (Section 4.2).
> - **S-PPO** addresses the second issue by incorporating clipping-bias normalization to adaptively downweight high-variance updates, applied to token-level PPO.
> - **ST-PPO** combines both mechanisms—turn-level importance sampling and clipping-bias correction—yielding the most stable behavior across model sizes and tasks.
>
> **Relationship between S-PPO and ST-PPO:**
> We emphasize that our goal is not to claim that **ST-PPO** is definitively superior to **S-PPO**. As shown in Figure 4, both variants exhibit similar and consistently strong convergence, with differences being task-dependent rather than statistically conclusive. Therefore, we present S-PPO and ST-PPO as complementary stabilized variants that both successfully stabilize multi-turn PPO training, rather than competing methods.
>
> **Summary of contributions:**
> Importantly, all three variants (Turn-PPO, S-PPO, and ST-PPO) represent distinct methodological contributions of this work, each addressing specific instability sources identified in our analysis. Empirically, all three demonstrate substantial improvements over baseline token-level PPO in terms of both training stability (lower gradient variance) and final task performance.
>
> To clarify the technical distinctions, the following table summarizes the key modifications and specific issues each variant addresses.
>
> | Method      | Importance Sampling (IS) Granularity | Credit Assignment Granularity | Gradient Stability Mechanism |
> |-------------|--------------------------------|-------------------------------|-------------------------------|
> | **Token-PPO** | Token-level IS ratio $w_t$ | Token-level credit | No bias correction |
> | **Turn-PPO**  | Turn-level IS ratio $w^{\text{turn}}_k$ | Aggregated turn-level credit  | No bias correction |
> | **S-PPO**     | Token-level IS ratio $w_t$ | Token-level credit | Gradient normalization via $\|C_{\text{token}}(\theta)\|_2^{-1}$ |
> | **ST-PPO**    | Turn-level IS ratio $w^{\text{turn}}_k$ | Aggregated turn-level credit  | Gradient normalization via $\|C_{\text{turn}}(\theta)\|_2^{-1}$ |
>
> As shown, each variant incrementally addresses specific sources of instability: Turn-PPO aligns credit assignment with task structure, S-PPO adds gradient stabilization at the token level, and ST-PPO combines both mechanisms.
>
> ---
>
> Next, we address the specific comments raised by each reviewer. We deeply appreciate the thoughtful engagement from all four reviewers and ACs throughout the reviewing process.
>
> Sincerely,
>
> **Authors of Submission 13970**

---

### Meta-Review · Area_Chair_hL6m · 2026-01-08

**Summary:**

The paper tackles an important instability problem in multi-turn PPO for LLMs and proposes a well-motivated stabilization mechanism, but its empirical evidence does not convincingly demonstrate that turn-level importance sampling provides meaningful benefits beyond clipping-bias correction alone.

**Reviewer Concerns:**

The central weakness of the paper is that the empirical evidence does not clearly justify the necessity or added value of the turn-level component, relative to the clipping-bias correction alone.

**Reviewer Scores:**

Reviewers are unanimous in suggesting a rejection for the paper, and it appears unlikely that a majority of them would raise their score to flip the final decision

---

### Decision · Program_Chairs · 2026-01-26

Reject